# Estimation of raindrop size distribution and rain rate with infrared surveillance camera in dark conditions

Jinwook Lee[1], Jongyun Byun[1], Jongjin Baik[1], Changhyun Jun[1], Hyeon-Joon Kim[1]

[1]Department of Civil and Environmental Engineering, College of Engineering, Chung-Ang University, Seoul, 06974, South Korea

*Correspondence to*: Hyeon-Joon Kim (hjkim22@cau.ac.kr)

**Abstract.** This study estimated raindrop size distribution (DSD) and rainfall intensity with an infrared surveillance camera in dark conditions. Accordingly, rain streaks were extracted using a *k*-nearest neighbor (KNN)-based algorithm. The rainfall intensity was estimated using DSD based on physical optics analysis. The estimated DSD was verified using a disdrometer for the two rainfall events. The results are summarized as follows. First, a KNN-based algorithm can accurately recognize rain streaks from complex backgrounds captured by the camera. Second, the number concentration of raindrops obtained through closed-circuit television (CCTV) images had values between 100 $mm^{-1}m^{-3}$ and 1,000 $mm^{-1}m^{-3}$, the root mean square error (RMSE) for the number concentration by CCTV and PARticle SIze and VELocity (PARSIVEL) was 72.3 $mm^{-1}m^{-3}$ and 131.6 $mm^{-1}m^{-3}$ in the 0.5 to 1.5 mm section. Third, the maximum raindrop diameter and the number concentration of 1 mm or less produced similar results during the period with a high ratio of diameters of 3 mm or less. Finally, after comparing with the 15-min cumulative PARSIVEL rain rate, the mean absolute percent error (MAPE) was 49% and 23%, respectively. In addition, the differences according to rain rate can be found that the MAPE was 36% at a rain rate of less than 2 mm $h^{-1}$ and 80% at a rate above 2 mm $h^{-1}$. Also, when the rain rate was greater than 5 mm $h^{-1}$, MAPE was 33%. We confirmed the possibility of estimating an image-based DSD and rain rate obtained based on low-cost equipment during dark conditions.

## 1 Introduction

Precipitation data is vital in water resource management, hydrological research, and global change analysis. The primary means of measuring precipitation is to use a rain gauge (Allamano et al., 2015) to collect raindrops from the ground. Due to the restrictions on the installation environment of the rain gauge, it can be difficult to understand the spatial rainfall distribution in mountains and urban areas (Kidd et al., 2017). Furthermore, the tipping-bucket-type rain gauge, which accounts for most rain gauges, has a discrete observation resolution (0.1 or 0.5 mm) for the discrete time steps, producing uncertainty in temporal rainfall variation. For this reason, weighing gauges are nowadays used very often instead of tipping-bucket-type. the weighing gauge is a meteorological instrument used to observe and analyze various precipitation, including rainfall and snowfall. Also, the tipping bucket has a large error due to the observation time delay when the rainfall is less than 10 mm $h^{-1}$ compared to the

weighing gauge. However, when the observation time size is set to 10 to 15 minutes, the relative percentage error has a very
low value of -6.7 to 2.5%, resulting in high accuracy (Colli et al., 2014).
In contrast, it is possible to obtain spatial rainfall information on a global scale with remote sensing techniques (Famiglietti
et al., 2015). However, remote sensing techniques provide only indirect measurements that must be continuously calibrated
and verified through ground-level precipitation measurements (Michaelides et al., 2009). Recently, a disdrometer capable of
investigating the microphysics characteristics of rainfall has been used for observation instead of the traditional rainfall
observation instrument (Kathiravelu et al., 2016). However, these devices cannot be widely installed because of their high cost
and difficulty in accessing observational data. Consequently, a high-resolution and low-cost ground precipitation monitoring
network has not yet been established.
With the advent of the Internet of Things (IoT) era, using non-traditional sources is attractive for improving the
spatiotemporal scale of existing observation networks (McCabe et al., 2017). In recent years, such cases have been common
in rainfall observation. For example, there have been attempts to estimate rainfall using sensors to capture signal attenuation
characteristics in commercial cellular communication networks (Overeem et al., 2016), vehicle wipers (Raibei et al., 2013),
and smartphones (Guo et al., 2019). Furthermore, crowdsourcing information has been used to confirm the utility of estimating
regional rainfall (Haberlandt and Sester, 2010; Rabiei et al., 2016; Yang and Ng, 2017).
In a similar context, a surveillance camera is a sensor with high potential. Surveillance cameras are often referred to as
closed-circuit television (CCTV). Compared with other crowdsourcing methods, the visualization data of surveillance cameras
are highly intuitive (Guo et al., 2017). Therefore, they have been used in various fields (Cai et al., 2017; Nottle et al., 2017;
Hua, 2018). In Korea, public surveillance camera installations have been rapidly increasing, from approximately 150,000 in
2008 to 1.34 million in 2020—approximately a public CCTV camera per 0.07 km$^2$. Thus, the potential for precipitation
estimation using camera sensing is expected to be greater in Korea.
Recently, various studies have been conducted to estimate rainfall intensity using the rain streak image obtained from
surveillance camera videos. Many studies attempted to use artificial intelligence to capture changes in the image captured by
the camera when it rains (Zen et al., 2019; Avanzato and Beritelli, 2020; Wang et al., 2022). In contrast, some studies have
tried to estimate rainfall intensity using geometrical optics and photographic analyses. Typically, the rain streak layer is
separated from the raw image or video. A rain streak is the visual appearance of raindrops caused by visual persistence—
raindrops falling because of the blur phenomenon of raindrop movement from the camera's exposure time appears as streaks
on the image. Garg and Nayar (2005) made one of the first attempts to measure this rainfall.
These previous studies indeed confirmed the possibility of rainfall measurement using surveillance cameras. However,
several limitations still prevent the actual expansion of the measurement systems using surveillance cameras. In general, most
surveillance cameras are installed for monitoring purposes, and people's faces are inevitably captured. Therefore, it is not easy
to disclose the data due to privacy concerns. Data storage and transmission are also limitations. Since most surveillance
cameras use a hard disk, data must be taken out directly. In other words, rainfall estimation cannot be done in real-time unless
a system is in place to transmit data over the Internet. In addition, the applicability to night-time is more limited. In the case of

general surveillance cameras in the past, observation is possible only when sunlight exists. For the observation system to expand, these various limitations must be addressed, and it seems that a lot of time and effort are needed. Nevertheless, research to develop algorithms using surveillance cameras in various conditions and to confirm applicability can have sufficient meaning. The case of dark conditions is one of the conditions worth studying. This is because the recently installed surveillance cameras are equipped with an infrared recording function, so most cameras will be able to take videos at night soon. However, the final purpose of utilizing these devices and the method is not to replace existing devices. It could be a supplement to improve the spatiotemporal resolution and accuracy of existing observation instruments. In particular, a study on the drop size distribution of rainfall, rather than simple rainfall estimation, would have more potential application value.

Since then, many studies have been conducted to develop and improve efficient algorithms. Allamano et al. (2015) proposed a framework to estimate the quantitative rainfall intensity using camera images based on physical optics from a hydrological perspective. Dong et al. (2017) proposed a more robust approach to identifying raindrops and estimating rainfall using a grayscale function, making grayscale subtraction nonlinear. Jiang et al. (2019) proposed an algorithm that decomposes rain-containing images into rain streak layers and rainless background layers using convex optimization algorithms and estimates instantaneous rainfall intensity through geometric optical analysis.

Some studies (e.g., Dong et al., 2017) have sought to estimate raindrop size distribution (DSD) using a surveillance camera. However, the existing studies have focused on the time when video can be captured with visible light. It is impossible to obtain input data without visible light using the existing image-based rainfall measurement method. Thus, these methodologies are only applicable in daytime conditions. However, when recording using infrared rays, it is possible to obtain a rainfall image even when there is no sunlight. No study has estimated the rain in dark conditions to our knowledge. Furthermore, most previous studies did not verify the estimated DSD using a disdrometer. In contrast, this study estimated DSD with an infrared surveillance camera in dark conditions, based on which rainfall intensity was also estimated. Rain streaks were extracted using a k-nearest neighbor (KNN)-based algorithm. The DSD was used to calculate rainfall intensity with physical optics analysis and verified using a PARticle SIze and VELocity (PARSIVEL) disdrometer (Löffler-Mang and Joss, 2000).

## 2 Methodology

### 2.1 Recording video containing rain streaks using an infrared surveillance camera

The surveillance camera records video. The video looks continuous, but it is also composed of discrete still images, so-called frames. The frequency of recording frames (i.e., acquisition rate) is called frame per second (fps). In other words, fps is how many images are taken per second for recording video. Another important factor in video recording is exposure time. Exposure time, also called shutter speed, refers to the time the camera sensor is exposed to light to capture a single frame. The real raindrops are close to a circle, but in a single image, the raindrops look like a streak. This is because raindrops move at a high speed during the exposure time. Therefore, the raindrops that moved during the exposure time are visualized in the rain streaks in a single frame.

Fig. 1 shows an example of capturing a raindrop for a single frame. Here, only the raindrops near the point of focus are
visible, and objects that are more than a certain distance appear invisible. That is, the point where the focus is best is called the
focus plane, and there is a range in which it can be recognized that objects are focused before and after the focus plane. The
closest plane that can be considered to be in focus is called the near-focus plane, and the farthest plane is called the far-focus
plane. This range is generally called depth of field (DoF). Ultimately, the rainfall intensity can be estimated based on the
volume and raindrops in the DoF.
In this study, an infrared surveillance camera was considered under dark conditions. Here, the dark condition refers to a
condition in which raindrops cannot be captured by a general surveillance camera with visible light. Infrared cameras emit
near-infrared rays through an infrared emitter and receive the reflected light from the objects. Accordingly, it has the advantage
of being able to detect raindrops that are invisible to the human eye.

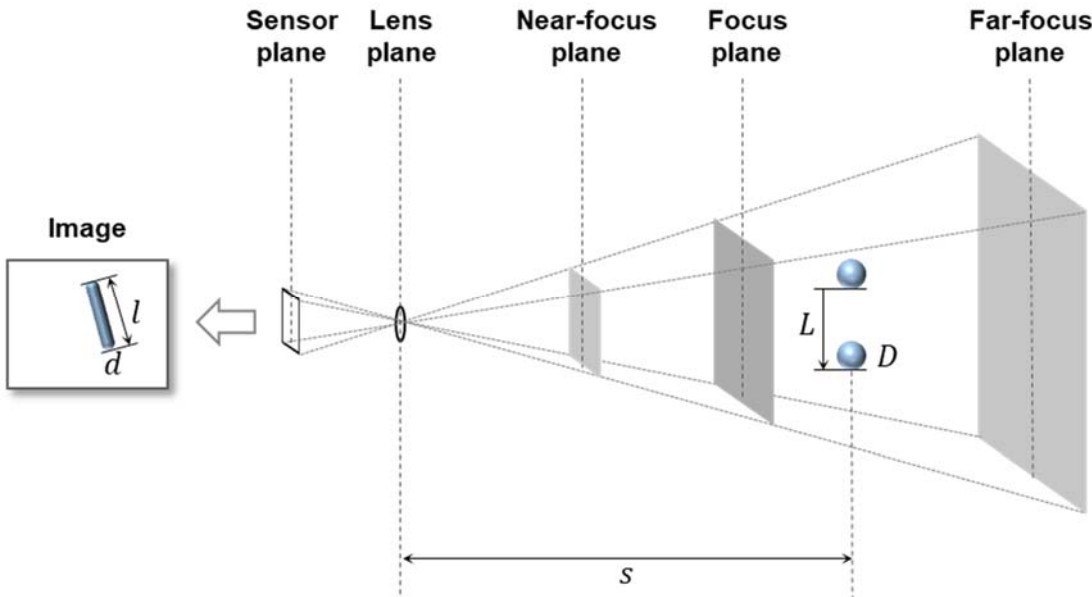

**Figure 1. Schematic diagram of the photographed rain streak in the image and the movement of a raindrop during the exposure**
**time.**

### 2.2 Algorithm for identifying rain steaks and estimating DSD and rain rate

Image-based rainfall estimation can be divided into two processes: identifying rainfall streaks and estimating DSD. Fig. 2
illustrates these processes in a flowchart. Identifying rain streaks requires an algorithm that separates the moving rain streaks
from the background layer. Next, in estimating DSD, raindrops are extracted from the image of the rain streaks, and the overall
distribution is obtained.

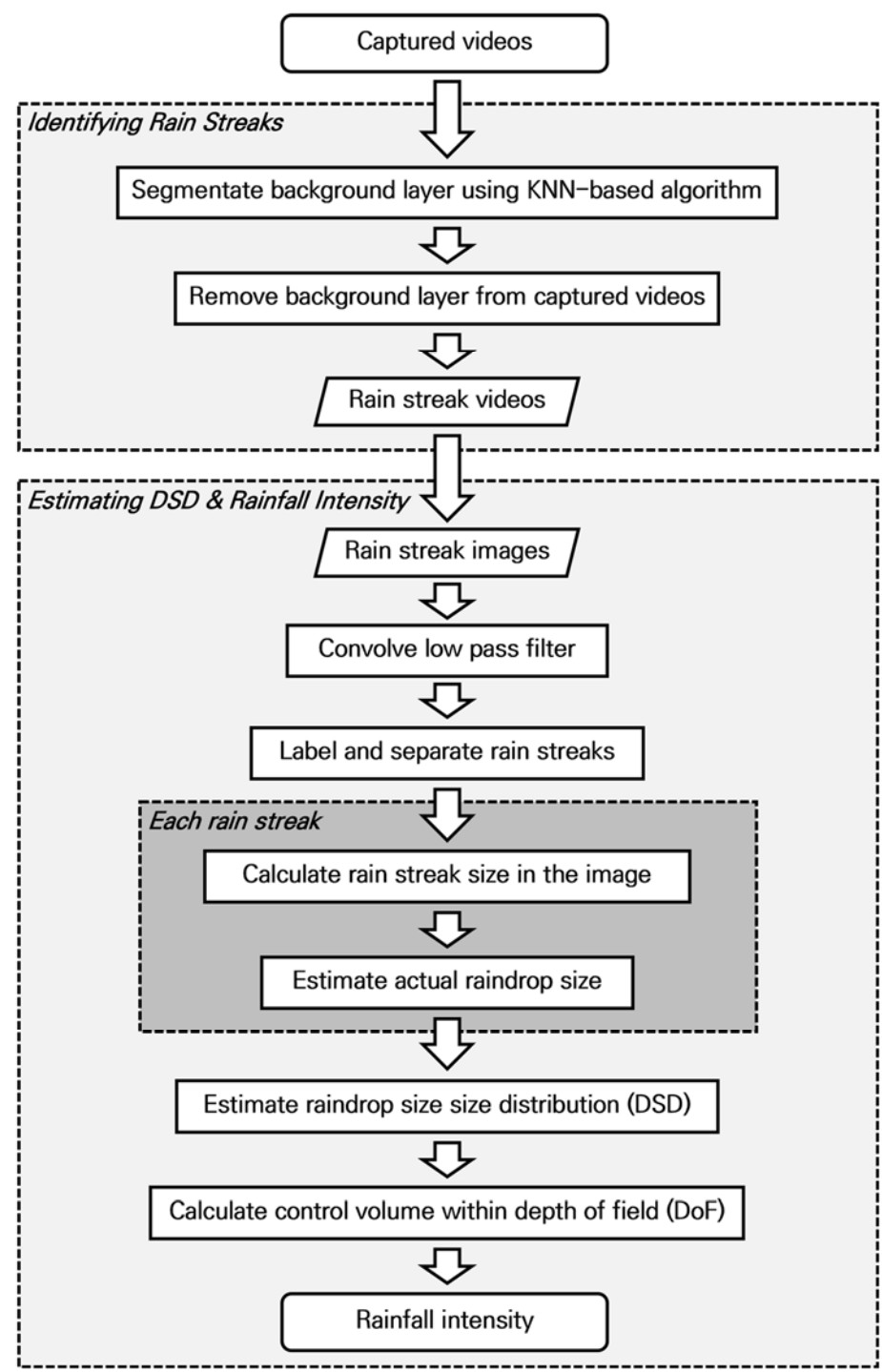

**Figure 2. Flowchart of the methodology for estimating DSD and rainfall intensity.**
Most existing algorithms aim to remove raindrops in images because raindrops are considered noise in object detection
and tracking (Duthon et al., 2018). Such algorithms are categorized into multiple-image-based and single-image-based
approaches (Jiang et al., 2018).
For example, Garg and Nayar (2007) classified the conditions in which the brightness difference between the previous
pixel and that of the next pixel exceeds a specific threshold over time, assuming that the background is fixed. Improved
algorithms were then developed considering the temporal correlation of raindrops (Kim et al., 2015) and chromatic properties
(Santhaseelan and Asari, 2015). Tripathi and Mukhopadhyay (2014) proposed a framework that removes rain that reduces the
visibility of the scene to improve the detection performance of image feature information. However, single-image-based
algorithms rely more on the properties of raindrops (Deng et al., 2018). The central idea of a single-image-based algorithm is
to decompose rain-containing images into rainless layers (Li et al., 2016; Deng et al., 2018; Jiang et al., 2018).
An image including grayscale rainfall may be mathematically expressed in a two-dimensional (2D) matrix in which each
element has a grayscale value. A single image (m×n) is expressed as follows (Jiang et al., 2018):
$$O = B + R, \tag{1}$$
where $O \in R^{m \times n}$, $B \in R^{m \times n}$, and $R \in R^{m \times n}$ are the raw image, rain-free background layer, and rain streak layer.
Accordingly, various algorithms are available for rain streak identification. Different still image and video-based
algorithms have been proposed to eliminate objects such as moving objects for application to actual surveillance cameras.
However, most of these algorithms face optimization problems because of the vast number of decision variables (Jiang et al.,
2019). This task is not easy to solve or requires excessive computation time. Therefore, existing studies present techniques
suitable for post-analysis rather than application in real-time. The use of complex algorithms can increase versatility and
accuracy, but there is a trade-off that reduces computational speed. The time required for such computing is a critical
disadvantage in practical applications for estimating rainfall intensity.
In this study, a KNN-based segmentation algorithm (Zivkovic and Heijden, 2006), a popular non-parametrical method for
background subtraction, was considered for segmenting the rain streaks (foreground) and background layers. KNN is used in
classification and regression problems (Bouwmans et al., 2010). The concept of KNN is that similar things are close—the
KNN-based segmentation algorithm finds the closest k samples (neighbors) to the unknown sample using Euclidean distance
to determine the class (i.e., foreground or background). Thus, the KNN-based segmentation method to detect foreground
changes in the video was used to identify rain streaks by recording infrared videos under conditions with little background
influence. In the algorithm, The KNN subtractor works by updating the parameters of a Gaussian mixture model for more
accurate kernel density estimation (Trnovszký et al., 2017). KNN is more efficient for local density estimation (Qasim et al.,
2021); therefore, the algorithm is highly efficient if the number of foreground pixels is low.
We used the package provided by OpenCV to implement the KNN-based segmentation algorithm (Zivkovic and Heijden,
2006). Accordingly, three main parameters (history, dist2Threshold, detectShadows) needed to be set. Table 1 presents the
description of the parameters used for the KNN background subtractor package.
**Table 1. Parameters in KNN background subtractor package in OpenCV.**

| Parameter | Description |
|---|---|
| history | Length of the history |
| dist2Threshold | Threshold on the squared distance between the pixel and the sample to decide whether a pixel is close to that sample. This parameter does not affect the background update. |
| detectShadows | If true, the algorithm will detect shadows and mark them. This decreases the speed slightly, so if you do not need this feature, set the parameter to false. |

It is essential to capture raindrops within the camera's depth of field (DoF) to calculate the final DSD and rainfall intensity.
Accordingly, this study proposed a novel algorithm to extract each rain streak from the rain streaks image. First, we applied a
low-pass filter to the rain streaks image to remove unfocused raindrops that may remain in the image, which smooths each
pixel using a 2D kernel. Videos from infrared mode have usually a blur effect. Thus, the additional 2D kernel was applied to
remove the pixels having blur. Highly detailed parts (e.g., out-of-focus raindrops and some noises) are erased, leaving some
clear rain streaks. A background layer with a value of 0 and a part not in the image were separated to extract the rain streaks
and labeled one by one to identify each rain streak from the image.
Because the rain streak observed in the surveillance camera image causes an angle difference (influenced by the wind), a
diameter estimation process considering the angle of the rain streak (fall angle of a raindrop) is required. If the angle of rain
steak is considered and converted to the raindrop diameter through the horizontal pixel size in the image, the shape change in
the raindrop because of air buoyancy (i.e., during the falling of the raindrop) may not be reflected, and overestimation can
occur.
Accordingly, the representative angle of each extracted rain streak was calculated. The border information of each rain
streak was obtained, and the center axis information of the rain streak was obtained based on the border information to calculate
the drop angle. Moreover, the rain streak was rotated to set the long and short axes of the streak at 0° and 90°, using the angle
information.
The size of raindrops in the rain streaks image can be estimated through the analysis of microphysical characteristics of the
raindrop and geometric optical analysis (Keating, 2002). The instantaneous velocity of a raindrop on the ground can be
estimated from the exposure time and the size of the raindrop. However, the distance from the raindrop to the lens surface (i.e.,
the object distance) is unknown and should be inferred. Object distance can be calculated through physical optics analysis
because it causes perspective distortion. Assuming a raindrop is spherical, the length of the trajectory where the raindrop falls
when the camera is exposed and the diameter of the raindrop can be inferred through the lens equation (Keating, 2002):
$$L(s) = \frac{d_f - f}{d_f \cdot f} \frac{h_s}{h_p} l_p s,$$ (2)
$$D(s) = \frac{d_f - f}{d_f \cdot f} \frac{w_s}{w_p} d_p s,$$ (3)

where $s$ is the distance from the raindrop to the lens plane (mm). $L(s)$ and $D(s)$ are the length of falling trajectory during camera exposure (rain streak) and the raindrop's diameter. $d_f$ is the focus distance (mm), and $f$ is the focal length (mm). $h_s$ and $w_s$ are the vertical and horizontal sizes of the active area of the image sensor (mm), and $h_p$ and $w_p$ are the vertical and horizontal sizes of the captured image (in the number of pixels). $l_p$ and $d_p$ are the length and width of the rain streaks in the image (in the number of pixels).

It is then possible to infer the falling speed of raindrops using the camera's exposure time (Jiang et al., 2019), as follows:

$$v(s) = \frac{L(s)}{1000\tau}, \tag{4}$$

where $\tau$ is the exposure time of the camera (seconds) and $v(s)$ is the fall velocity of the raindrop from the image. Furthermore, the fall velocity of a raindrop can be approximated by an empirical formula for raindrop diameter. The most frequently used equation is as follows (Atlas et al., 1973; Friedrich et al., 2013):

$$v(D) = 9.65 - 10.3\exp(-0.6D), \tag{5}$$

where $D$ is the raindrop diameter and $v$ is the fall velocity of the raindrop. The actual diameter of raindrops can be obtained by solving the equation with the fall velocity obtained through the exposure time and Eqs. (4) and (5). Furthermore, the DoF for the images using the camera's setting information can be calculated, and the effective volume for estimating rainfall intensity can be obtained. Details of the process are described in previous studies (Allamano et al., 2015; Jiang et al., 2019). The control volume must be determined to estimate the rainfall intensity using the diameter of each raindrop. An understanding of DoF is required to achieve the volume. The DoF is simply the range at which the camera can accurately focus and capture the raindrops. Calculating this range requires obtaining the near and far focus planes as follows:

$$s_n = \frac{d_f \cdot f^2}{f^2 + N \cdot c_p \cdot (d_f - f)}, \tag{6}$$

$$s_f = \frac{d_f \cdot f^2}{f^2 - N \cdot c_p \cdot (d_f - f)}, \tag{7}$$

where $s_n$ and $s_f$ are the distances from the near and far focus planes. $c_p$ is the maximum permissible circle of confusion, a constant determined by the camera manufacturers. $N$ is the F-number of the lens relevant to the aperture diameter. Accordingly, the theoretical sampling volume ($V$, m$^3$) indicates the truncated rectangular pyramid between the near and far focus planes:

$$V = \frac{1}{3 \cdot 10^9} \left(\frac{d_f - f}{d_f \cdot f}\right)^2 w_s h_s (s_f^3 - s_n^3), \tag{8}$$

Then, we used the gamma distribution equation, Eq. (6), proposed by Ulbrich (1983), to calculate DSD parameters using data at every 1 min interval.

$$N(D) = N_0 D^\mu \exp(-\Lambda D), \tag{9}$$

where $N(D)$ (mm$^{-1}$m$^{-3}$) is the number concentration value per unit volume for each size channel, and $N_0$ (mm$^{-1-\mu}$m$^{-3}$) is an intercept parameter representing the number concentration when the diameter has 0 value. $D$ (mm) and $\Lambda$ (mm$^{-1}$) are the drop diameter and slope parameter. Raindrops smaller than 8.0 mm were used to avoid considering non-weather data such as leaps and bugs (Friedrich et al., 2013).

The gamma distribution relationship is a function of formulating the number concentration per unit diameter and unit volume. It was proposed by Marshall and Palmer (1948) as an improved model of exponential distribution as a favorable form to reflect various rainfall characteristics. By including the term containing $\mu$ in the distribution function, the shape of the number concentration distribution for small drops smaller than 1 mm is improved.

$$N(D) = N_0 \exp(-\Lambda D), \tag{10}$$

As the $\Lambda$ decreases, the slope of the distribution shape decreases, and the proportion of the large drops increases. Conversely, as the value increases, the distribution slope becomes steeper, and the weight of the large particles decreases. When $\mu$ has a large value, the distribution is convex upward, and it has a distribution with a sharp decrease in number concentration at small diameters. Whereas when it has a negative value, the distribution is convex downward with an increase in the concentration of drops smaller than 1 mm. In the gamma distribution, the $\mu$ is mainly affected by the difference in concentration of raindrops smaller than 3 mm (Vivekanandan et al., 2004).

Vivekanandan et al. (2004) explained the reason for using the gamma distribution as follows. First, it is sufficient to calculate the rainfall estimation equation using only the first, third, and fourth moments (Eq. (11)) (Smith, 2003). Second, the long-term raindrop size distribution has an exponential distribution shape (Yuter and Houze, 1997).

The raindrop size distribution observed from the ground is the result of the microphysical development of raindrops falling from precipitation clouds. The drop size distribution shape is changed during fall by microphysical processes such as collision, merging, and evaporation, and changes in the concentration of drops larger than 7.5 mm and small drops occur mainly. As a result, the drop size distribution observed on the ground mainly follows the gamma distribution shape (Ulbrich, 1983; Tokay and Short, 1996). The gamma distribution relationship should be used to analyze the distribution of raindrops that are actually floating and falling.

$$M_n = \int_{D_{min}}^{D_{max}} D^n N(D) dD, \tag{11}$$

Eq. (11) indicates a moment expression for the $n^{th}$ order. For example, the second moment is calculated as the product of the square of the diameter of each channel and the number concentration and the diameter of each channel. Each moment value has a different microphysical meaning. Therefore, the gamma distribution including three dependent parameters is more advantageous in reflecting the microphysical characteristics of the precipitation system than the exponential distribution including two dependent parameters. Eq. (11) can be expressed in gamma distribution format as follows:

$$M_n = \int_{D_{min}}^{D_{max}} D^n N(D) dD = N_0 \Lambda^{-(\mu+n+1)} \Gamma(\mu+n+1), \tag{12}$$

where $N_T$ (total number concentration, m$^{-3}$) is the zero-order moment ($M_0$) and represents the total number concentration of
raindrops per unit volume. $\eta$ was determined for calculating $\mu$ and $\Lambda$. In this study, a combination of moments in the ratio of
$M_2$, $M_4$, and $M_6$, which accurately represents the characteristics of small raindrops, was applied (Vivekanandan et al., 2004):
$$\eta = \frac{\langle M_4 \rangle^2}{\langle M_2 \rangle \langle M_6 \rangle} = \frac{(\mu+3)(\mu+4)}{(\mu+5)(\mu+6)}, \tag{13}$$
$\mu$ and $\Lambda$ are calculated as follows:
$$\mu = \frac{(7-11\eta)-[(7-11\eta)^2-4(\eta-1)(30\eta-12)]^{1/2}}{2(\eta-1)}, \tag{14}$$
$$\Lambda = \left[\frac{M_2\Gamma(\mu+5)}{M_4\Gamma(\mu+3)}\right]^{1/2} = \left[\frac{M_2(\mu+4)(\mu+3)}{M_4}\right]^{1/2}, \tag{15}$$
A larger value of $D_m$ (mm) estimated using Eq. (16), the diameter of the average mass of raindrops contained in the unit
volume, indicates that predominantly larger drops are distributed.
$$D_m = \frac{M_4}{M_3}, \tag{16}$$
$R$ (mm h$^{-1}$) is the rain rate calculated using Eq. (17).
$$R = \frac{6\pi}{10^4} \int_{D_{min}}^{D_{max}} D^3 N(D) V(D) dD, \tag{17}$$

## 3 Study site and observation equipment

This study used a building's rooftop as the study site. The building is the Chung-Ang University's Bobst Hall, located in the
central region of Seoul in Korea. It is located at 37° 30' 13" north latitude and 126° 57' 27" east longitude, at an elevation of
42 m. Fig. 3 illustrates the CCTV (marked with a red circle) and PARSIVEL installed at the study site. The CCTV was used
for the main analysis, and PARSIVEL was considered for verification purposes.

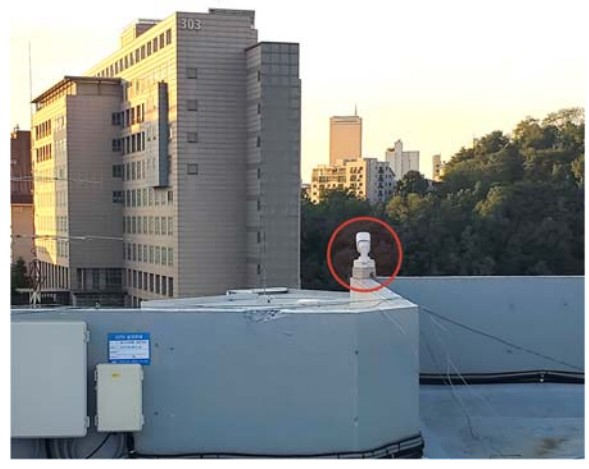 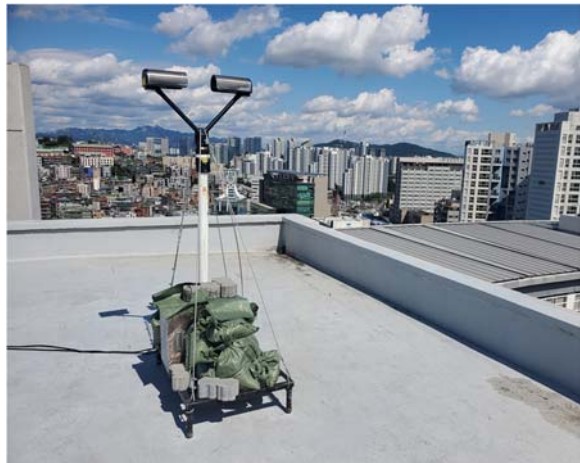

(a) Surveillance camera                         (b) PARSIVEL

**Figure 3. Observation measurements considered in this study.**

The CCTV model used in this study is DC-T333CHRX, developed by IDIS. The camera has a 1/1.7 inch complementary

metal-oxide semiconductor (CMOS) with a height and width of 5.70 mm and 7.60 mm. The focal length is 4.5 mm, and the
F-number of the lens is 1.6. The shutter speed was set to 1/250 s, and the frame per second (fps) was set to 30. The infrared
ray distance is 50 m. The maximum permissible circle of confusion is 0.005 mm. The camera's resolution is 1,080 pixels for
the height and 1,920 pixels for the width, but the cropped images (640×640 pixels) were considered for the analysis.

The PARSIVEL is a ground meteorological instrument that can observe precipitation particles' diameter and fall speed

(e.g., raindrops, snow particles, hail). The meteorological information, including raindrop size, is used to estimate the
quantitative precipitation amount and reveal the precipitation system's microphysical characteristics and development
mechanism.

The PARSIVEL used in this study is the second version of the instrument manufactured by OTT in Germany, and it is

improved the observation accuracy of small particles. The PARSIVEL uses a laser-based optical sensor to send a laser from
the transmitter and continuously receive it from the receiver (Fig. 4). As the laser beam moves from the transmitter to the
receiver, the precipitation particle passes over the laser beam, and the size and velocity of the precipitation particle are observed
(Nemeth and Hahn, 2005). The diameter and velocity of the particle are calculated by calculating the time the particle passes
through the laser and the laser intensity that decreases during the passage (Fig. 5).

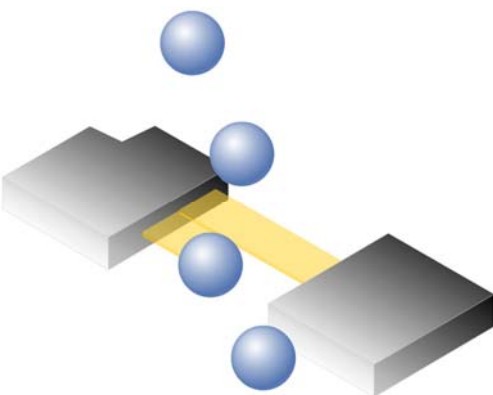

**Figure 4. Functional principle of the PARSIVEL disdrometer.**

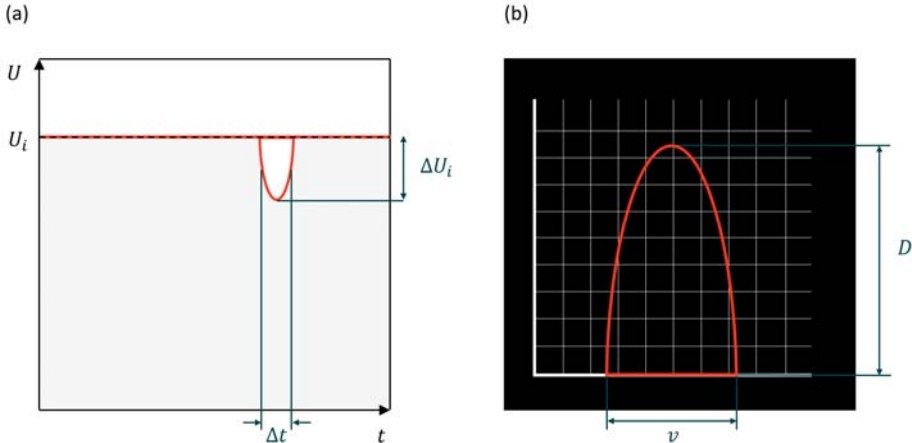

**Figure 5. (a) Signal changes whenever a particle falls through the beam anywhere within the measurement area. (b) The degree of**
**dimming is a measure of the particle's size; together with the duration of the signal, the fall velocity can be derived.**
Parameters such as rain rate, reflectivity, and momentum of raindrops are calculated through particle concentration values
for each diameter and falling speed channel obtained through PARSIVEL observation. In this study, the temporal resolution
of the observation data was set to 1 minute. The particle diameters from 0.2 to 25 mm (Table 1 in Appendix) and fall velocity
from 0.2 to 20 m s$^{-1}$ (Table 2 in Appendix) can be observed by the PARSIVEL. The particle diameter and the fall speed each
have 32 observation channels, so the number of observed particles for the time resolution set in 1,024 channels (32×32) is
observed. The first and second channels of diameter are not included in the observable range of the PARSIVEL and are treated
as noise. Therefore, the observation data of the first and second diameter channels were not considered in the actual analysis.
The detailed information on the specifications of the PARSIVEL is presented in Table 2.
**Table 2. Technical information of the PARSIVEL disdrometer.**

| Wavelength of optical sensor | | 780 nm |
|---|---|---|
| Measuring area | | $30 \times 180$ mm (54 cm$^2$) |
| Measuring range | Size | $0.2 \sim 25$ mm (32 channel class) |
| | Fall velocity | $0.2 \sim 20$ m s$^{-1}$ (32 channel class) |
| Precipitation intensity | | $0.001 \sim 1,200$ mm h$^{-1}$ |
| Measurement time interval | | 10 sec $\sim$ 60 min |
| Instrument dimensions (H×W×D) | | $670 \times 600 \times 114$ mm |

## 4 Application result

### 4.1 Rainfall event

We considered two rainfall events from 1945 LST on March 25, 2022, to 0615 LST on March 26, 2022 (case 1), and 2100 LST on September 5, 2022, to 0300 LST on September 6, 2022 (case 2). Fig. 6 illustrates the hyetographs of the rainfall event considered in this study according to the time resolution. The total rainfall of case 1 and 2 is 19.5 and 48.7 mm based on the PARSIVEL, respectively. The maximum rain rate is 10.0 and 20.7 mm h$^{-1}$ based on the 1 min resolution, and 5.0 and 14.5 mm h$^{-1}$ based on the 15 min resolution for case 1 and case 2.

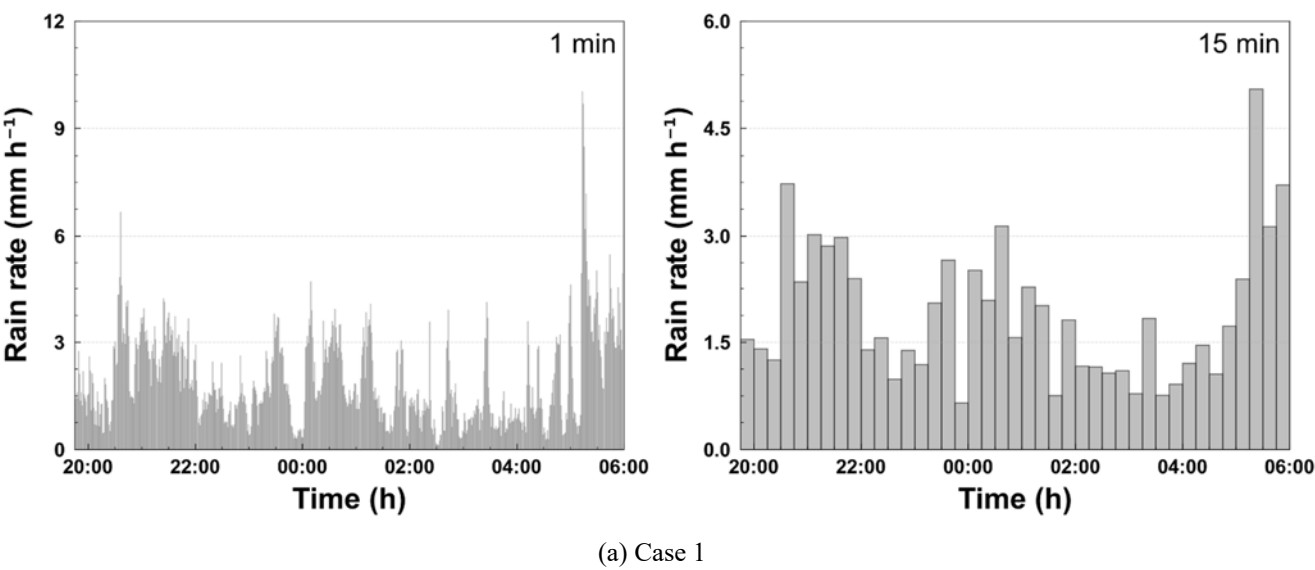

(a) Case 1

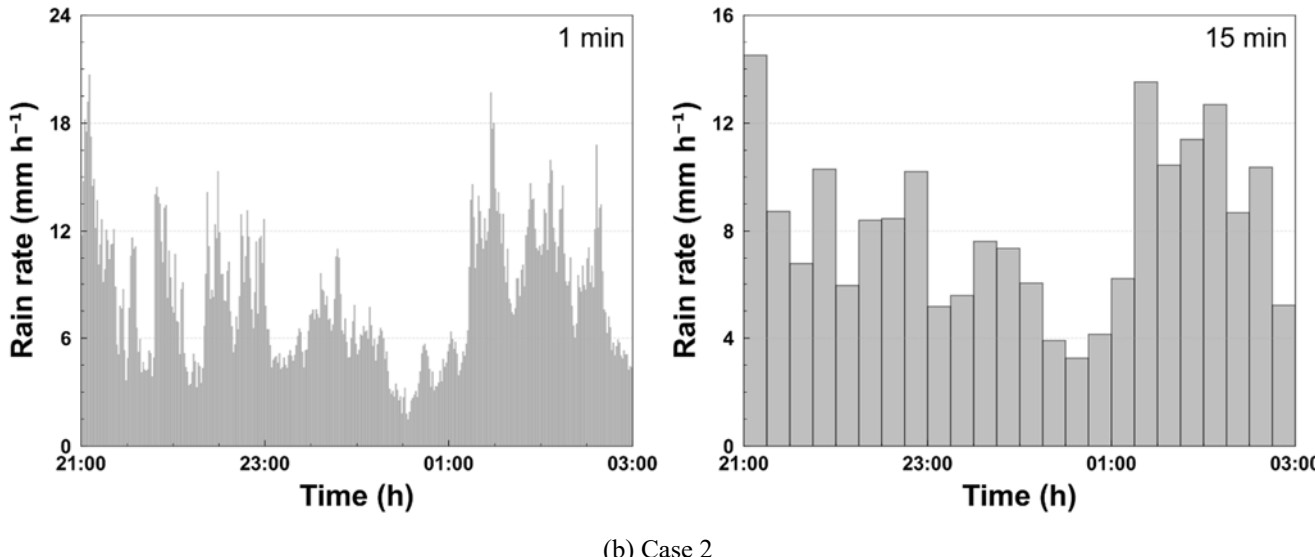

(b) Case 2

**Figure 6. Hyetograph of PARSIVEL and rain gauge observation data for the rainfall events considered in this study (left: 1 min resolution, right: 10 min resolution).**

In order to secure the quantitative reliability of the PARSIVEL observation data, rain gauge observation data were used to verify the rainfall calculated through the PARSIVEL observation. The rainfall data used for verification are rain gauge observation data operated by KMA (Korea Meteorological Administration) installed closer than 4 km from the PARSIVEL observation site (Table 3). The rainfall comparison period is from September 14, 2021, to October 4, 2022, including the period of the analysis case. Fig. 7 shows scatter plots comparing hourly rain rates from rain gauges and PARSIVEL. As a result of comparison with the observation data at three rain gauge sites, it had low MAE (Mean Absolute Error), RMSE (Root Mean Square Error), MAPE (Mean Absolute Percent Error) values of less than 0.11 mm h$^{-1}$, 0.6 mm h$^{-1}$, and 8%. Also, correlation values were more than 0.9.

**Table 3. Location information of rain gauge observation sites.**

| Rain gauge site | Latitude (°) | Longitude (°) | Range from PARSIVEL site (km) |
|---|---|---|---|
| G1 | 37.4933 | 126.9175 | 3.73 |
| G2 | 37.5196 | 126.9763 | 2.42 |
| G3 | 37.5249 | 126.9390 | 2.87 |

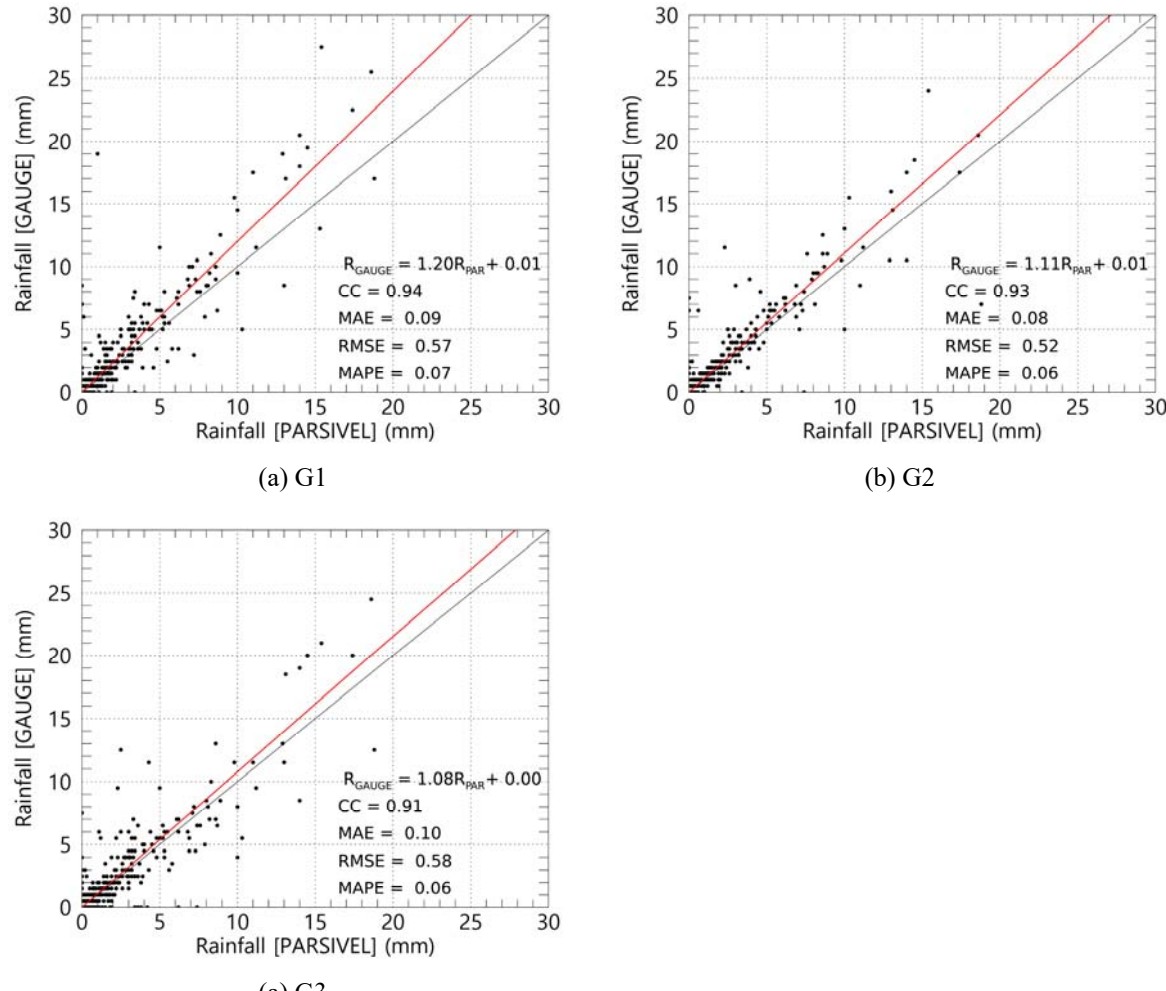

(a) G1  (b) G2

(a) G3

**Figure 7. Scatter plot of rainfall amount every 1 hour from the PARSIVEL observation and the rain gauge observation.**

## 4.2 Identifying rainfall streaks

The rain streaks were distinguished from the original raw images using the KNN-based algorithm described in Section 2.2. Accordingly, two parameters (history and dist2Threshold) were set to default values (500 and 400). The other parameter (detectShadows) was set to "false." Fig. 8 illustrates the raw, background, and rain streaks images for an example time image (20:30:57 March 25, 2022), scaled in yellow to make it easier to verify the visual change.

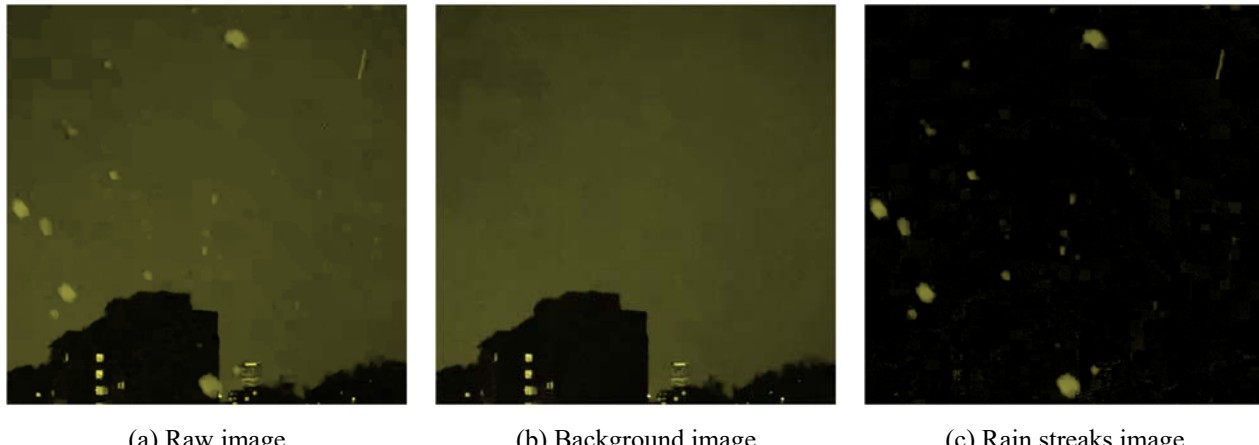

| (a) Raw image | (b) Background image | (c) Rain streaks image |

**Figure 8. Segmentation example of a raw image into background and rain streaks image based on KNN-based algorithm (20:30:57**
**March 25, 2022).**
As confirmed in Fig. 8, adequate background separation performance can be achieved using the KNN-based method used
in this study. Because it is an infrared camera and the camera's exposure time is 1/250 s, the length of rain streaks is relatively
short. The longer the exposure time, the longer the raindrops appear on the image (Schmidt et al., 2012; Allamano et al., 2015).
If the exposure time is too long, some rain streaks may penetrate the image. In this case, it is difficult to estimate the rain streak
length, a clue for estimating raindrop size.
The identification algorithm was implemented using Anaconda Software Distribution on a workstation with an AMD Ryzen
5 5600X 6-Core Processor and 32 GB RAM. The computing time for the 15 min video was approximately 50 s using only
CPU computation. As described previously, the KNN-based algorithm used in this study has high-speed computing
performance compared with various algorithms based on optimization, so it will likely have an advantage in real-time
applications.
**4.3 Estimation of DSD and rain rate**
The rain streaks image presented in Fig. 8(c) was not considered for the final DSD estimation because of noise and factors
other than rain caused by the sudden brightness change. As described in Section 3, a low-pass filter was first applied to the
rain streaks image.
The 10×10 kernel was applied considering the total image size (640×640), and each grid value of the kernel was set to
0.01. The set kernel was filtered by convolution pixel by pixel. Moreover, the convolution was performed once more using the
following 2D kernel [0 1 0; -1 0 1; 0 -1 0] to highlight the rim of the rain streaks. A background layer with a value of 0 and a
part not in the image were separated to extract the rain streaks, which were labeled one by one to identify each rain streak from
the image. Fig. 9(a) illustrates the example result after performing the processes described above in Fig. 8(c). Each rain streak
was then separated and labeled, as in Fig. 9(b).

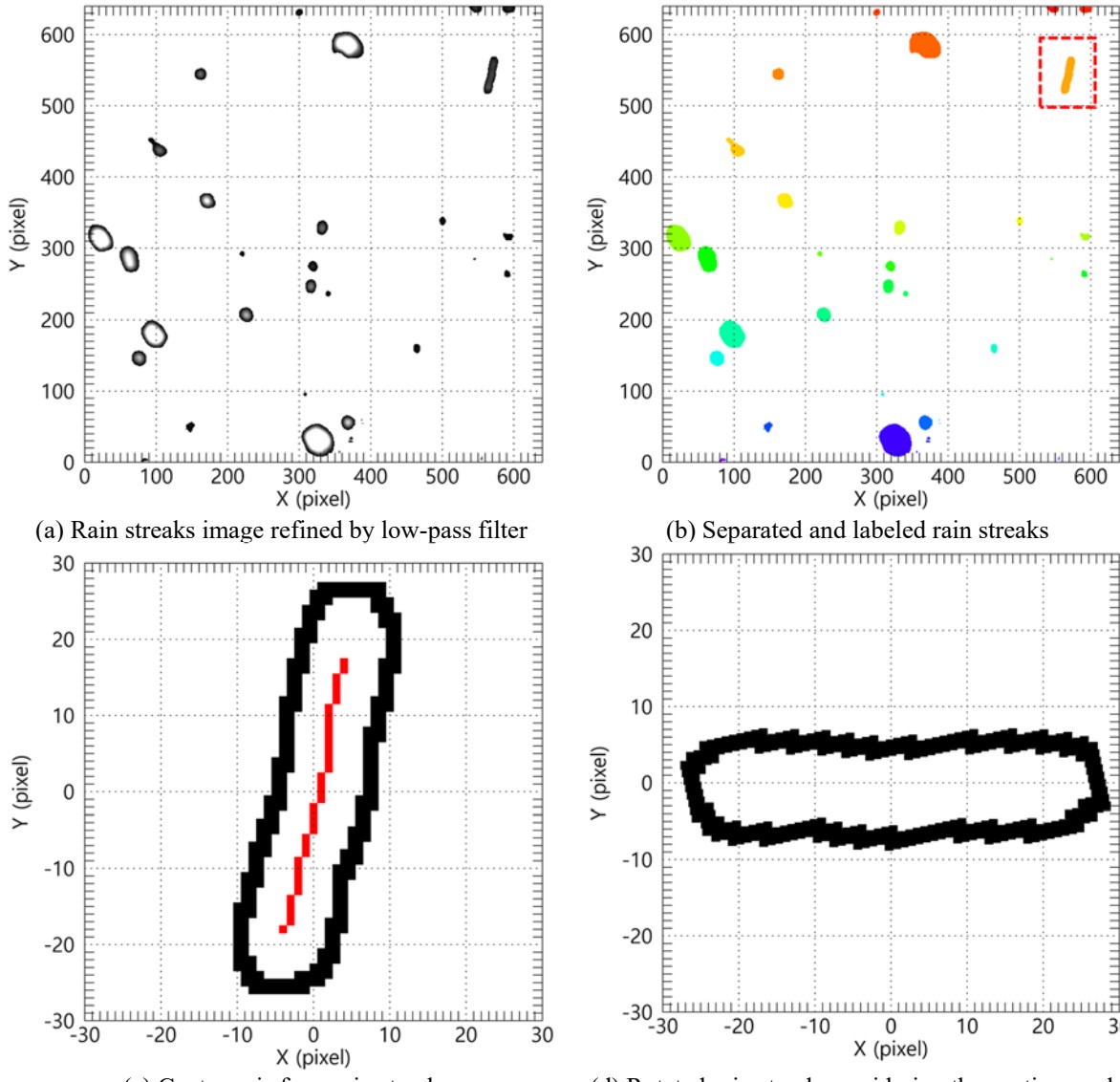

(a) Rain streaks image refined by low-pass filter  (b) Separated and labeled rain streaks

(c) Center axis for a rain streak  (d) Rotated rain streak considering the canting angle

**Figure 9. Extraction example of rain streak based on the proposed algorithm.**
The border information of each rain streak needed to be obtained. The center axis was calculated by connecting the center
(median) of the minimum pixel and maximum pixel values of the x-axis for each y-axis using border information. The angle
of rain steak was obtained from the slope value obtained by calculating the linear function through the center axis's x and y
pixel number values. Fig. 9(c) is an example of the extraction of a rain streak extracted from the image of Fig. 9(b).
The drop angle was then calculated, and the rain streak was rotated using the angle information. Raindrops can be broken
up by strong wind or collisions between raindrops during falling. The maximum difference value between the minimum and
maximum pixel number values of the y-axis calculated using border information of the rotated rain steak was used to calculate
the raindrop diameter and exclude the influence of the distorted shape of rain steak by break up (Fig. 9(d)) (Testik, 2009;
Testik and Pei, 2017). Fig. 9(d) illustrates the result of the final process. If the rain streaks overlap, the diameter of the raindrops
can be estimated as large. To reduce the overestimation of raindrop diameter, this study tried to find the main central axis
coordinates of overlapping rain streaks and set the longest central axis as the representative value. Then, estimate the primary
diameter by calculating the distance between each pixel value of the set central axis and the edge pixels of rain streaks.

Fig. 10 illustrates the time series of the number concentration and $D_m$ obtained from CCTV and PARSIVEL. From 1945

LST to 2350 LST, the maximum number concentration of lower than 1,000 $mm^{-1}m^{-3}$ was observed from the PARSIVEL
observation, and from 2000 LST to 2010 LST, a number concentration lower than 100 $mm^{-1}m^{-3}$ was observed. At 2005 LST,
large raindrops (of 3.8 mm) were observed, resulting in a sharp increase in $D_m$ above 2 mm. In contrast, in the results based on
CCTV images, the number concentration of less than 10,000 $mm^{-1}m^{-3}$ was continuously demonstrated during the entire analysis
period, and a number concentration greater than 5,000 $mm^{-1}m^{-3}$ was observed before 2200 LST. Because the proportion of
small drops was high, $D_m$ was predominantly less than 1.5 mm.

From 0000 LST to 0100 LST, both CCTV and PARSIVEL-based data had a predominant maximum diameter of about 2.4

mm. At 0035 LST, raindrops larger than 3.2 mm were observed in PARSIVEL, but raindrops less than 3 mm were not observed
in CCTV. However, the number concentration of small diameters of 0.5 mm or less had similar values between 1,000 and
5,000 $mm^{-1}m^{-3}$. Despite the difference in the maximum size of the drops, there was no predominant difference in the $D_m$
because the number concentration of raindrops smaller than 1 mm had similar values.

From 0300 LST to 0530 LST, number concentrations higher than 5,000 $mm^{-1}m^{-3}$ in the raindrops smaller than 1 mm were

observed using PARSIVEL. However, CCTV data revealed that number concentrations less than 5,000 $mm^{-1}m^{-3}$ were
consistently observed. From 0500 LST to 0510 LST, CCTV image-based number concentration consistently appeared as about
1.2 mm, whereas $D_m$ was smaller than 0.7 mm in PARSIVEL. The cause for the rapid decrease in $D_m$ of the PARSIVEL was
that the CCTV-based maximum diameter is about 2.4 mm, which was similar to the PARSIVEL observation data, but the
number concentration of 0.5 to 0.6 mm raindrops observed by PARSIVEL had a large value of more than 10,000 $mm^{-1}m^{-3}$.


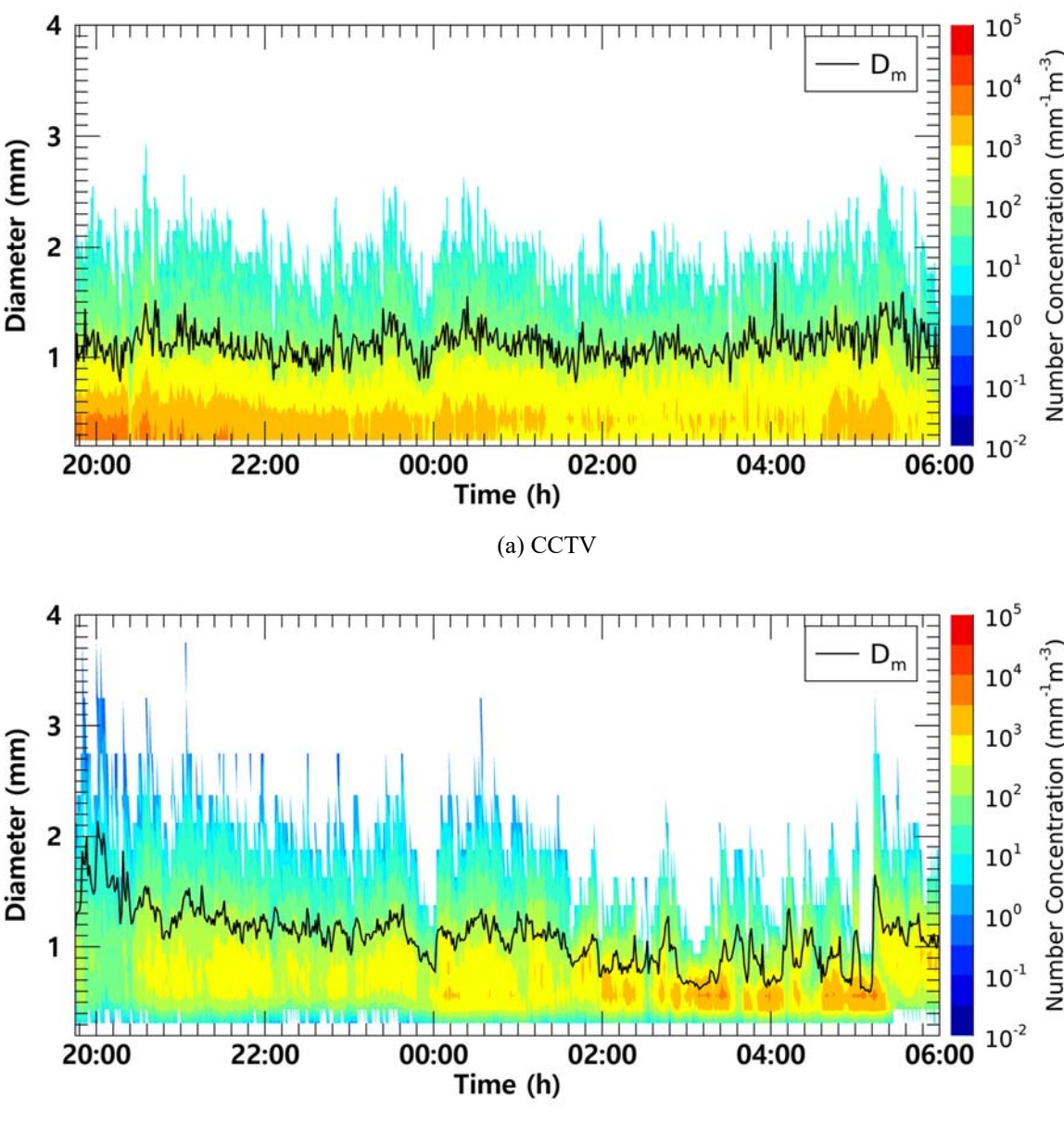

(a) CCTV

(b) PARSIVEL

**Figure 10. Time series of number concentration and D$_m$ (black coloured line) from (a) the surveillance camera images, (b) the PARSIVEL observation data from 2145 LST on March 25 to 0600 LST on March 26, 2022 (case 1).**

Fig. 11 illustrates the average number concentration versus diameter of raindrops calculated using CCTV image and PARSIVEL observation data from 1945 LST on March 25 to 0600 LST on March 26, 2022. The PARSIVEL disdrometer data has a fixed raindrop diameter channel; thus, it can differ in number concentration depending on the diameter channel setting.

Therefore, in this study, the simulated DSD through the gamma model was also analyzed to compare the distribution of rainfall
particles.
For raindrop diameters from 0.7 to 1.5 mm, the simulated and observed number concentrations produced similar values.
However, above 1.5 mm, the model-based number concentration was under-simulated. From these results, in the precipitation
case selected in this study, the gamma model appears limited in simulating the number concentration of raindrops larger than
3 mm. In diameters from 0.2 to 1.0 mm and above 1.5 mm, the number concentration obtained from CCTV images tended to
be higher than that from PARSIVEL observation. PARSIVEL observation data decreased sharply for diameters smaller than
0.3 mm. In contrast, CCTV gradually increased the number concentration as the diameter decreased.

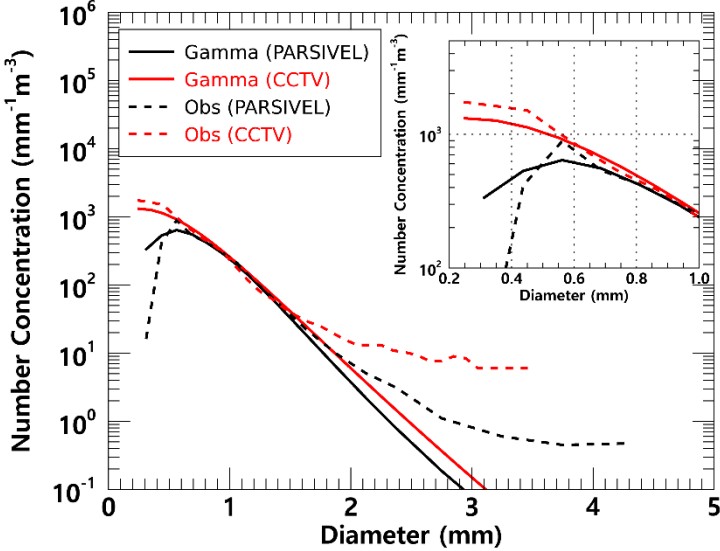

**Figure 11. Average number concentration versus diameter from the surveillance camera images and the PARSIVEL (case 1).**
Rainfall intensity was estimated based on the obtained number concentration from CCTV images and PARSIVEL. The
near ($s_n$) and far ($s_f$) focus planes were calculated as 718 and 1,648 mm from Eqs. (8) and (9). The DoF was calculated as 930
mm. The focal distance was set to 1 m, referring to previous studies (Dong et al., 2017; Jiang et al., 2019). The control volume
was 2.9 m$^{-3}$, applying Eq. (10) with the variables determined above. Fig. 12 illustrates the rain rate time series calculated using
CCTV images and PARSIVEL observation data. The increase or decrease in rain rate according to time change based on
CCTV data followed the trend of rainfall intensity change based on PARSIVEL observation data.
At 2037 LST, the PARSIVEL-based rain rate was 5.9 mm h$^{-1}$, but the CCTV-based rain rate was overestimated to be
higher than 10 mm h$^{-1}$. On the other hand, the CCTV-based rain rate was underestimated by about 2 mm h$^{-1}$ than the
PARSIVEL-based rain rate at 0514 LST. Quantitative changes in CCTV-based rain rate showed a similar tendency to increase
and decrease the number concentration of raindrops smaller than 1 mm and the maximum diameter. From 0100 LST to 0200
LST, when the number concentrations of CCTV and PARSIVEL had similar values, the rain rate also showed similar results.

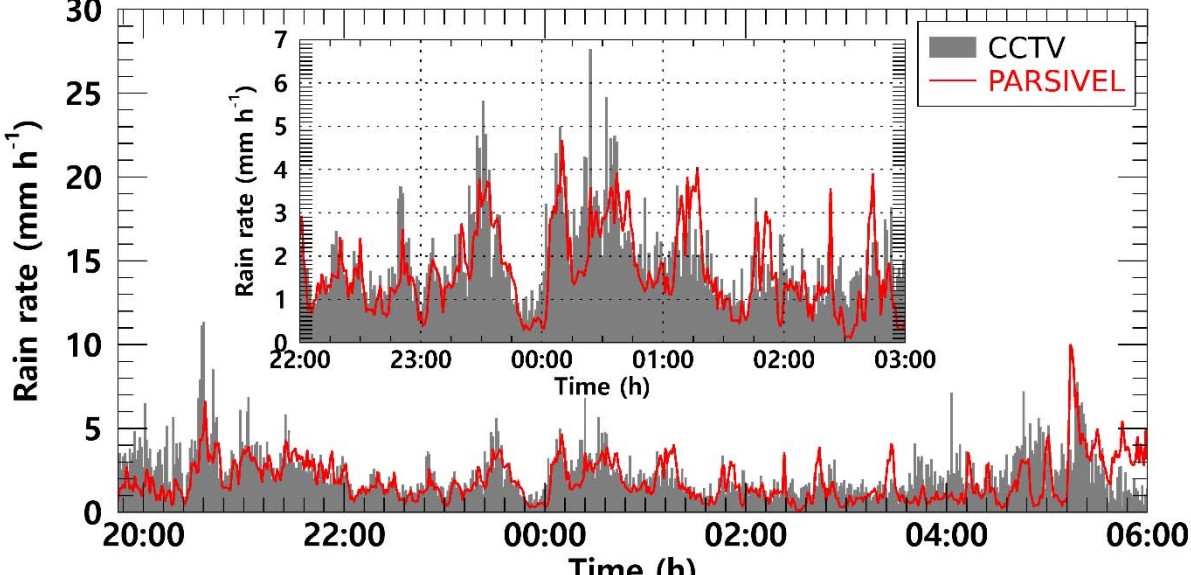

**Figure 12. The rain rate time series calculated from the surveillance camera images (gray bar) and PARSIVEL observation data**
**(red line) from 2145 LST on March 25 to 0600 LST on March 26, 2022 (case 1).**
Fig. 13 illustrates the scatter plot of the average rain rate every 15 min from the PARSIVEL observation and the CCTV
images. Uncertainty exists in the resolution of the rain gauge in the 1 min step. Accordingly, the time step for analysis is set
to 15 min. The slope of the regression line was 0.71 because the CCTV-based rain rate tended to be overestimated at a rain
rate of weaker than 2 mm h$^{-1}$.
The cumulative average rainfall intensity every 15 min was weaker than 10 mm h$^{-1}$, concentrated at a rain rate less than 6
mm h$^{-1}$, so the correlation coefficient (CC) was 0.64. Furthermore, the MAE, RMSE, and MAPE were 0.61 mm h$^{-1}$, 0.99 mm
h$^{-1}$, and 48%. Differences according to rain rate can also be determined. The accuracy is higher at a rain rate smaller than 2
mm h$^{-1}$ as a boundary. The MAE, RMSE, and MAPE were 0.29 mm h$^{-1}$, 0.72 mm h$^{-1}$, and 38% for a rain rate of 2 mm h$^{-1}$ or
less, and 0.58 mm h$^{-1}$, 1.17 mm h$^{-1}$, and 55% for a rain rate above 2 mm h$^{-1}$.
The statistical values of the rain rate and DSD parameters for the rainfall cases analyzed in this study are summarized in
Table 3. The rain rate and $D_m$ calculated using CCTV images were 0.459 mm h$^{-1}$ and 0.025 mm more than the values calculated
using PARSIVEL observation data on average, respectively. A high rain rate and $D_m$ were caused by overestimating the number
concentration for raindrops larger than 1.5 mm confirmed in Fig. 10. The number concentration for the small diameter (less
than 0.3 mm) was higher in the CCTV data than in the PARSIVEL data. Due to the high concentration value of the number
concentration of raindrops below 0.5 mm and above 2 mm, the CCTV-based rain rate had a large value.
In the $D_m$ calculated through the PARSIVEL observation data, the concentration change of small drops over time was large,
and the variance (0.063 mm) of $D_m$ was large due to the rapid change in number concentration. The variability of the maximum
diameter was greater in the PARSIVEL observation data, but the variance of the rain rate was greater in the CCTV data. The
large variability of the concentration of raindrops below 3 mm affected the change in the rain rate. Also, due to the high number
concentration of small drops, the skewness of the CCTV (1.903) based rain rate had a higher value than that of the PARSIVEL
(1.589) based rain rate. The low variability (0.063 mm) of the $D_m$ calculated from CCTV data means that the change in the
shape of the raindrop size distribution was small, supported by the low variance of $\Lambda$ (3.016 mm$^{-1}$).
**Table 4. Statistical values of the rain rate and DSD parameters for case 1.**

|  |  | $R$ (mm h$^{-1}$) | $D_m$ (mm) | $\log_{10}N_0$ (mm$^{-1-\mu}$m$^{-3}$) | $\mu$ (unitless) | $\Lambda$ (mm$^{-1}$) |
|---|---|---|---|---|---|---|
| PARSIVEL | Mean | 1.905 | 1.091 | 7.379 | 7.394 | 11.829 |
|  | Variance | 1.667 | 0.063 | 15.170 | 35.975 | 88.288 |
|  | Skewness | 1.589 | 0.551 | 2.470 | 2.015 | 2.714 |
|  | Kurtosis | 5.189 | 1.233 | 7.751 | 5.132 | 9.165 |
| CCTV | Mean | 2.364 | 1.116 | 4.857 | 2.131 | 5.713 |
|  | Variance | 1.998 | 0.021 | 0.472 | 1.680 | 3.016 |
|  | Skewness | 1.903 | 0.536 | 1.109 | 0.628 | 1.151 |
|  | Kurtosis | 6.073 | 1.041 | 2.188 | 0.739 | 2.506 |


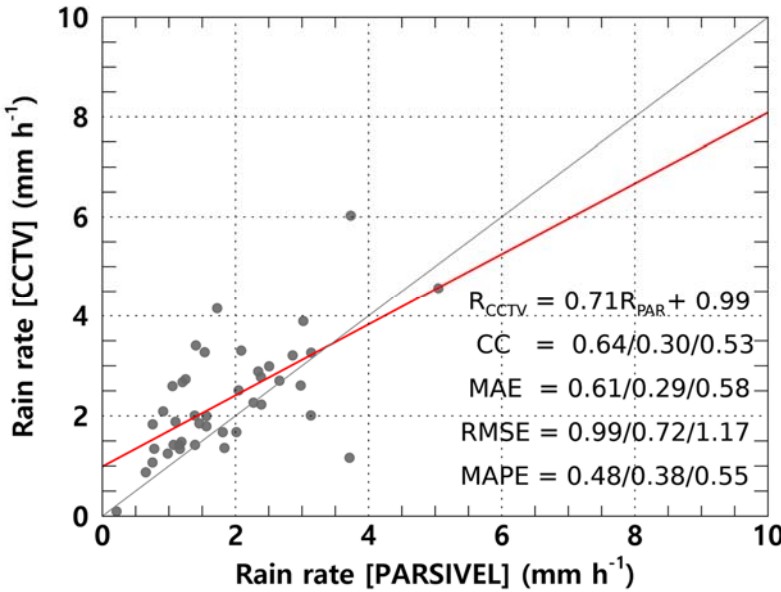

**Figure. 13. Scatter plot of average rain rate every 15 minutes from the PARSIVEL observation and the surveillance camera images**
**(case 1). Red line is linear regression. Scatter plot displays CC, MAE, RMSE, and MAPE for R > 0 mm h$^{-1}$, R < 2 mm h$^{-1}$, and R ≥ 2**
**mm h$^{-1}$ (sequentially from left to right).**

Fig. 14 illustrates the time series of the number concentration and $D_m$ obtained from CCTV and PARSIVEL for case 2. In

both CCTV and PARSIVEL observation data, the number concentration for a diameter between 0.5 mm and 1.5 mm had a
value between 500 mm$^{-1}$m$^{-3}$ to 5,000 mm$^{-1}$m$^{-3}$, and there was no significant change in the number concentration with time.

The maximum diameter also consistently had a value close to about 3 mm, and the $D_m$ was also similar to about 1.5 mm

because the maximum diameter and the number concentration of 1 mm intermediate drop had similar values.

From 0100 LST to 0230 LST, the maximum particle diameter through CCTV was overestimated, resulting in a large value

close to 3.5 mm. As a result, the $D_m$ value increased significantly to more than 2 mm. PARSIVEL data showed a sharp decrease
in the number concentration of 1 mm drops at 0030 LST, and an increase in $D_m$ under the influence of the decreased number
concentration. However, in the case of CCTV, only raindrops smaller than 1.5 mm were observed at the time, and there was a
similar decrease in $D_m$ (about 1.1 mm).

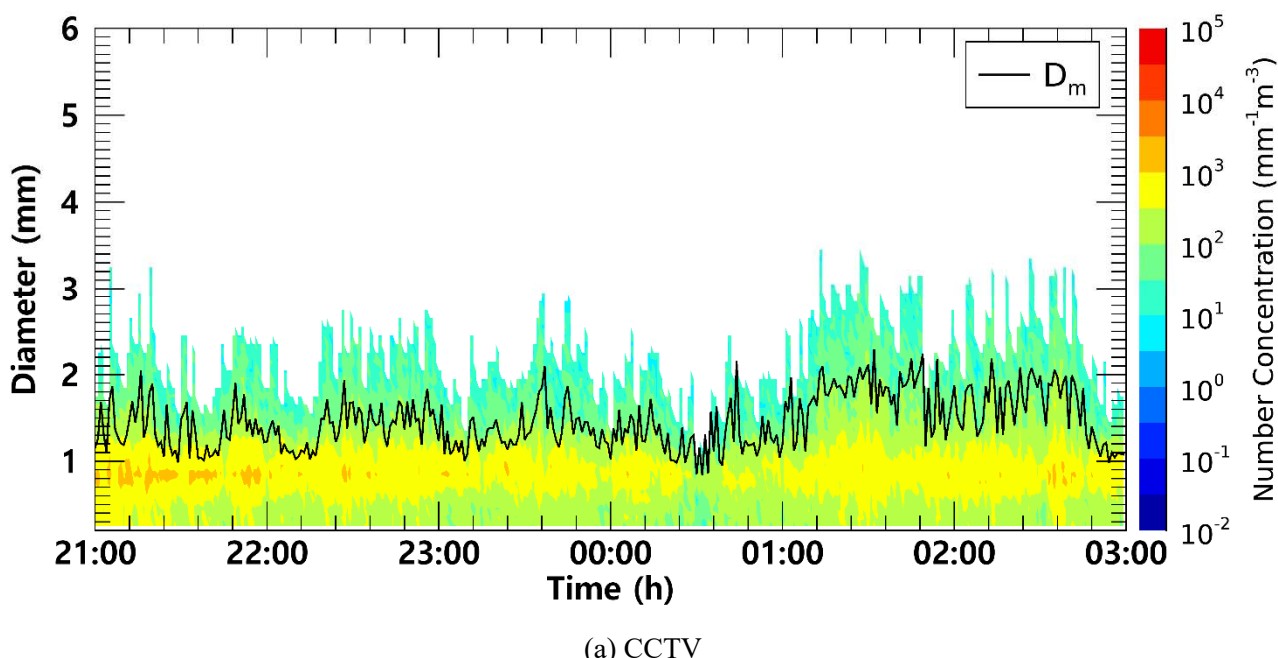

(a) CCTV

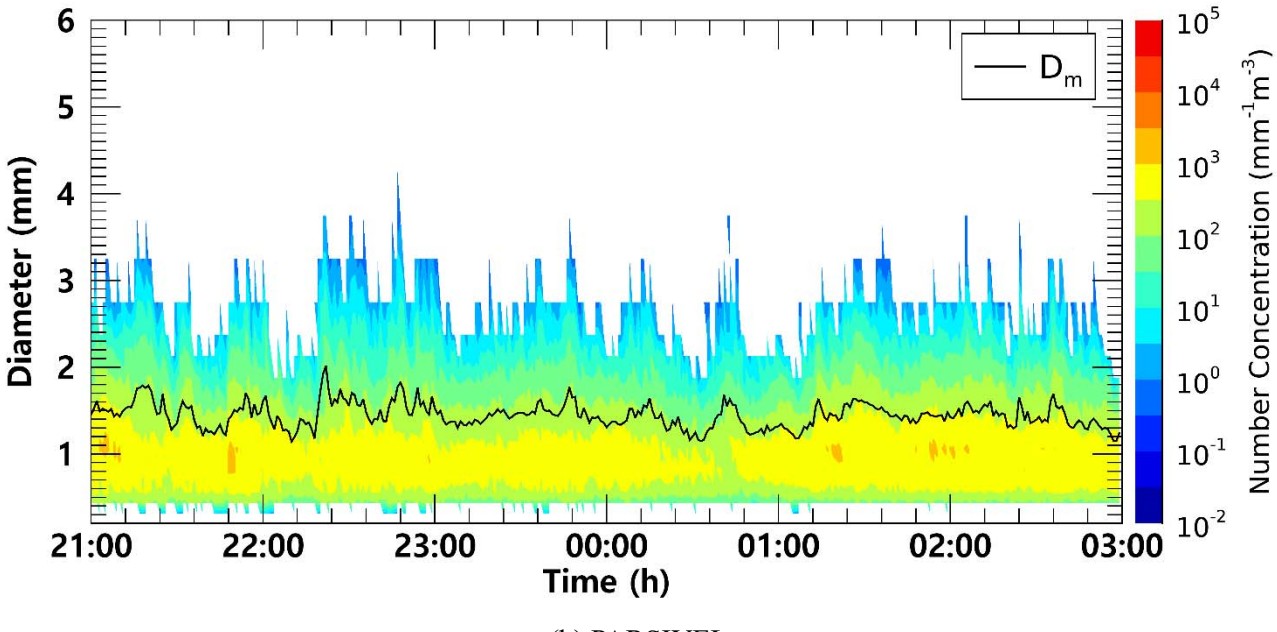

**Figure 14. Time series of number concentration and Dm (black coloured line) from (a) the surveillance camera images, (b) the PARSIVEL observation data from 2100 LST on September 5 to 0300 LST on September 6, 2022 (case 2).**

As clearly shown in Fig. 14, there was no significant difference in number concentration according to the time change. The average number concentration distribution also showed similar results because the number concentration values were concentrated at 1,000 $mm^{-1}m^{-3}$ concentration in both observation instruments. (Fig. 15). As in case 1, PARSIVEL observation data showed a tendency to underestimate in sections less than 0.5 mm and underestimated in sections larger than 2 mm compared to CCTV data. The diameter section where CCTV data is underestimated compared to PARSIVEL data was from 1 mm to 2 mm. Since the number concentration of the CCTV data was underestimated in this section, the rain rate based on the number concentration data was also underestimated compared to the rainfall intensity based on the PARSIVEL data.

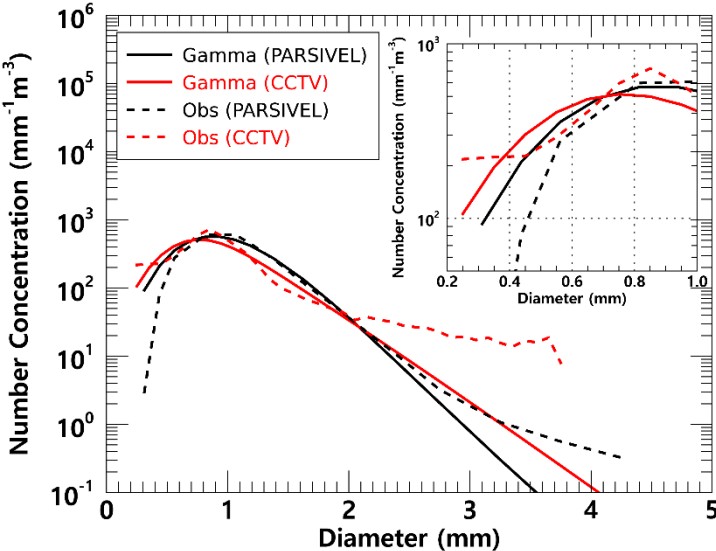

 **Figure 15. Average number concentration versus diameter from the surveillance camera images and the PARSIVEL (case 2).**

Between 2100 LST on September 5 and 0100 LST on September 6, when the number concentration of about 1 mm
raindrops is similar and the maximum diameter size is similar, the rain rate time series distribution has a value of about 5 mm
h[-1] and has a very similar flow. However, between 0130 LST and 0300 LST, which is a period with an overestimation of
raindrop diameter in CCTV observation data, the increase and decrease in rain rate were similar. However, the magnitude of
the increase and decrease in rain rate differed every 15 minutes. During that time, the maximum rain rate was less than 20 mm
h[-1] in the PARSIVEL observation data, while strong rainfall of 30 mm h[-1] or more was observed in the CCTV observation data.

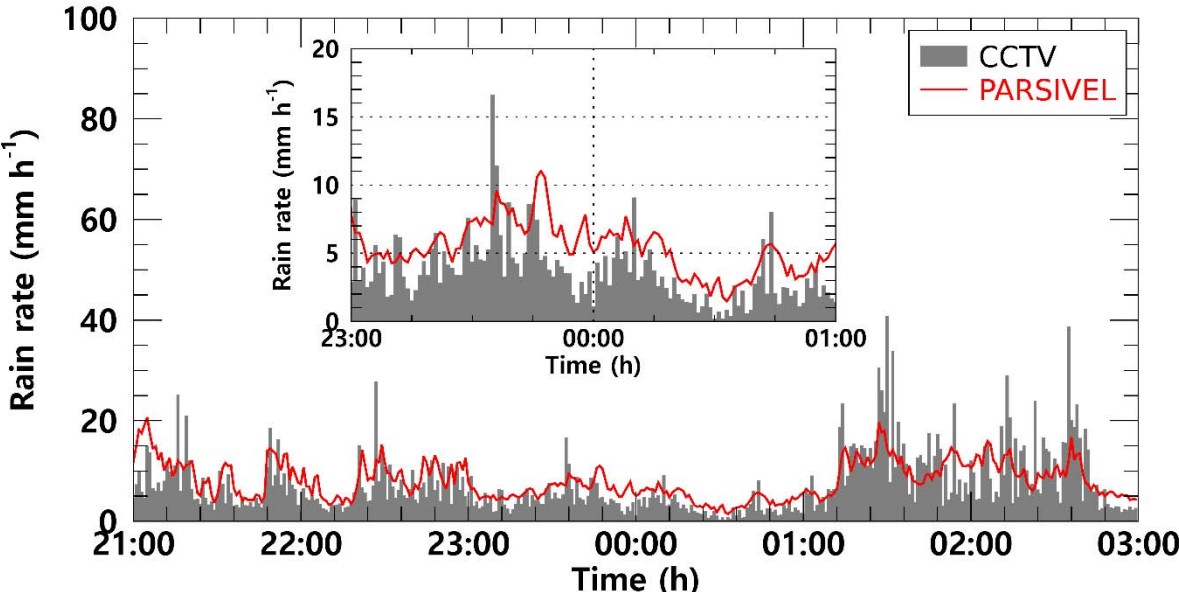

**Figure 16. The rain rate time series calculated from the surveillance camera images (gray bar) and PARSIVEL observation data (red line) from 2100 LST on September 5 to 0300 LST on September 6, 2022 (case 2).**

Fig. 17 illustrates the scatter plot of the average rain rate every 15 min from the PARSIVEL observation and the CCTV images for case 2. Compared to case 1, case 2 was a strong rainfall case with a rain rate of about 8.94 mm h$^{-1}$. Compared to the PARSIVEL observation data, the CCTV observation data showed a larger $D_m$ by 0.221 mm, while the $\log_{10}N_0$ showed a small feature of 1.1 mm$^{-1-\mu}$m$^{-3}$. As the weight of medium and large drops over 1 mm increased, $\mu$ and $\Lambda$ showed lower values of 4.262 and 5.397 mm$^{-1}$, respectively (Table 5). According to the 15-minute cumulative rain rate comparison result, the rain rate based on CCTV image data tends to be underestimated when it is less than 10 mm h$^{-1}$. Conversely, there was a tendency to overestimate the rainfall period of 10 mm h$^{-1}$ or more. This tendency was confirmed in case 1 which may be caused by recognizing overlapping rain streaks as a single big raindrop. MAPE had a low value of 0.3% or less regardless of the rain rate, and even though the rainfall intensity was relatively large compared to case 1, MAE and RMSE did not significantly increase. This is because there was no abnormally large value of CCTV rainfall during the rainfall period of case 2 compared to case 1.

**Table 5. Statistical values of the rain rate and DSD parameters for case 2.**

| | | $R$ (mm h$^{-1}$) | $D_m$ (mm) | $\log_{10}N_0$ (mm$^{-1-\mu}$m$^{-3}$) | $\mu$ (unitless) | $\Lambda$ (mm$^{-1}$) |
|---|---|---|---|---|---|---|
| PARSIVEL | Mean | 8.12 | 1.445 | 5.900 | 6.379 | 7.341 |
| | Variance | 13.82 | 0.020 | 1.160 | 6.498 | 5.596 |
| | Skewness | 0.65 | 0.447 | 1.061 | 0.9467 | 1.198 |
| | Kurtosis | -0.13 | 0.472 | 2.480 | 1.818 | 2.792 |
| CCTV | Mean | 8.94 | 1.666 | 4.813 | 4.262 | 5.397 |
| | Variance | 69.33 | 0.121 | 1.185 | 4.577 | 6.714 |
| | Skewness | 2.75 | 0.355 | 2.596 | 1.903 | 2.640 |
| | Kurtosis | 11.71 | -0.202 | 8.962 | 5.714 | 9.756 |

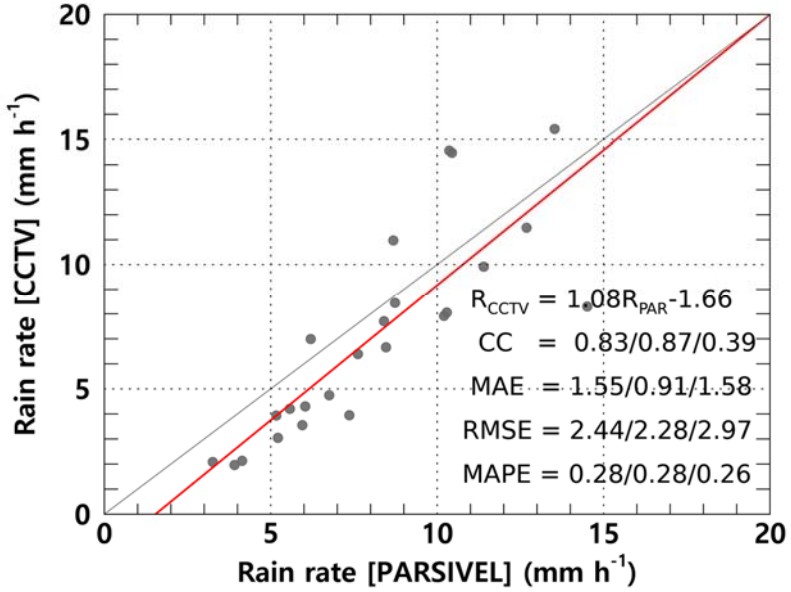

Fig. 17. Scatter plot of average rain rate every 15 minutes from the PARSIVEL observation and the surveillance camera images
(case 2). Red line is linear regression. Scatter plot displays CC, MAE, RMSE, and MAPE for R > 0 mm h⁻¹, R < 5 mm h⁻¹, and R ≥ 5
451  mm h⁻¹ (sequentially from left to right).

## 6 Conclusion

This study estimated DSD with an infrared surveillance camera, based on which rainfall intensity was also estimated. Rain streaks were extracted using a KNN-based algorithm. The rainfall intensity was estimated based on DSD using physical optics analysis. A rainfall event was selected, and the applicability of the method in this study was examined. The estimated DSD was verified using a PARSIVEL. The results from this study can be summarized as follows.

KNN-based algorithm illustrates suitable performance in separating the rain streaks and background layers. Furthermore, the possibility of separation for each rain streak and estimation of DSD was sufficient.

The number concentration of raindrops obtained through the CCTV images was similar to the actual PARSIVEL observed number concentration in the 0.5 to 1.5 mm section. In the small raindrops in the section of 0.4 mm or less, the PARSIVEL observation data underestimates the actual DSD. However, the CCTV image-based rain rate had an advantage over the raindrop-based data—the number concentration decreased rapidly as the number concentration gradually increased in the 0.2– 0.3 mm diameter section.

The maximum raindrop diameter and number concentration of less than 1 mm produced similar results during the period with a high ratio of diameters less than 3 mm. However, the number concentration was overestimated during the period when

raindrops larger than 3 mm were observed. The CCTV image-based data revealed that the rain rate was overestimated because of the overestimation of raindrops larger than 3 mm. After comparing with the 15-min cumulative PARSIVEL rain rate, the CCs—MAE, RMSE, and MAPE of case 1 (case 2)—were 0.61 mm h$^{-1}$ (1.55 mm h$^{-1}$), 0.99 mm h$^{-1}$ (1.43 mm h$^{-1}$), and 48% (44%). The differences according to rain rate can be identified. The accuracy is higher at a rain rate smaller than 10 mm h$^{-1}$ as a boundary.

The rain rate and $D_m$ calculated using CCTV images exhibited similar average values. The overestimated number concentration of 1.5 mm or larger caused high kurtosis for the rain rate and $D_m$ of CCTV-based data and a low $\mu$ value. Because of the high number concentration for raindrops larger than 3 mm of CCTV, the PARSIVEL observation data had a higher $\Lambda$ value than the result based on the CCTV data.

In this study, DSD was estimated using an infrared surveillance camera; the rain rate was also estimated. Consequently, we could confirm the possibility of estimating an image-based DSD and rain rate obtained based on low-cost equipment in dark conditions. Though the infrared surveillance camera considered in this study will not be able to replace traditional observation devices, if future studies can be continued to secure robustness, it will be an excellent complement to the existing observation system in terms of spatiotemporal resolution and accuracy improvement.

**Appendix. The diameter and fall velocity information for each diameter channel class.**

**Table 1. The representative diameter and spread for each diameter channel class.**

| Class number | Class average (mm) | Class spread (mm) | Class number | Class average (mm) | Class spread in (mm) |
|---|---|---|---|---|---|
| 1 | 0.062 | 0.125 | 17 | 3.250 | 0.500 |
| 2 | 0.187 | 0.125 | 18 | 3.750 | 0.500 |
| 3 | 0.312 | 0.125 | 19 | 4.250 | 0.500 |
| 4 | 0.437 | 0.125 | 20 | 4.750 | 0.500 |
| 5 | 0.562 | 0.125 | 21 | 5.500 | 1.000 |
| 6 | 0.687 | 0.125 | 22 | 6.500 | 1.000 |
| 7 | 0.812 | 0.125 | 23 | 7.500 | 1.000 |
| 8 | 0.937 | 0.125 | 24 | 8.500 | 1.000 |
| 9 | 1.062 | 0.125 | 25 | 9.500 | 1.000 |
| 10 | 1.187 | 0.125 | 26 | 11.000 | 2.000 |
| 11 | 1.375 | 0.250 | 27 | 13.000 | 2.000 |
| 12 | 1.625 | 0.250 | 28 | 15.000 | 2.000 |
| 13 | 1.875 | 0.250 | 29 | 17.000 | 2.000 |
| 14 | 2.125 | 0.250 | 30 | 19.000 | 2.000 |
| 15 | 2.375 | 0.250 | 31 | 21.500 | 3.000 |
| 16 | 2.750 | 0.500 | 32 | 24.500 | 3.000 |

**Table 2. The representative fall velocity and spread for each diameter channel class.**

| Class number | Class average (m s⁻¹) | Class spread (m s⁻¹) | Class number | Class average (m s⁻¹) | Class spread (m s⁻¹) |
|---|---|---|---|---|---|
| 1 | 0.050 | 0.100 | 17 | 2.600 | 0.400 |
| 2 | 0.150 | 0.100 | 18 | 3.000 | 0.400 |
| 3 | 0.250 | 0.100 | 19 | 3.400 | 0.400 |
| 4 | 0.350 | 0.100 | 20 | 3.800 | 0.400 |
| 5 | 0.450 | 0.100 | 21 | 4.400 | 0.800 |
| 6 | 0.550 | 0.100 | 22 | 5.200 | 0.800 |
| 7 | 0.650 | 0.100 | 23 | 6.000 | 0.800 |
| 8 | 0.750 | 0.100 | 24 | 6.800 | 0.800 |
| 9 | 0.850 | 0.100 | 25 | 7.600 | 0.800 |
| 10 | 0.950 | 0.100 | 26 | 8.800 | 1.600 |
| 11 | 1.100 | 0.200 | 27 | 10.400 | 1.600 |
| 12 | 1.300 | 0.200 | 28 | 12.000 | 1.600 |
| 13 | 1.500 | 0.200 | 29 | 13.600 | 1.600 |
| 14 | 1.700 | 0.200 | 30 | 15.200 | 1.600 |
| 15 | 1.900 | 0.200 | 31 | 17.600 | 3.200 |
| 16 | 2.200 | 0.400 | 32 | 20.800 | 3.200 |

**Data availability**


The raw videos and data used in the analysis can be downloaded from https://doi.org/10.6084/m9.figshare.c.6392430.v1, and
the sample codes are available in a public GitHub repository from https://github.com/jinwook213/Rain_CCTV.git.

**Acknowledgements**

This research was supported by the Korea Meteorological Administration Research and Development Program (KMI2022-
01910), by Basic Science Research Program through the National Research Foundation of Korea (NRF) funded by the Ministry
of Education (2022R1I1A1A01065554 and 2022R1A6A3A01087041), and by the Chung-Ang University Graduate Research
Scholarship in 2021.

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
