# Peer review of "Estimation of raindrop size distribution and rain rate with infrared surveillance camera in dark conditions"

_Atmospheric Measurement Techniques, 2022_

## Author Comment (AC1)

**Response Letter for Anonymous Referee #1**

**Overall Comments**

This manuscript provides an interesting new method to estimate microstructural and bulk rainfall properties from a CCTV camera. The idea is intriguing and the topic is fully appropriate for the journal. The manuscript needs overall a major and mandatory revision, as detailed in the following comments.

**Answer:**

We appreciate for your valuable comments and revision suggestions. As suggested by the referee, all the comments were considered in the revision of the manuscript.

Q1. Methodology section: the authors move forward to directly discuss and describe the algorithm. Important information on the measurement device is missing at this point, crucial for the scope of this journal. The reader at this point has the following questions: what are the technical characteristics of the camera? What are the actual input data? (description of the images, their resolution, acquisition rate, discussion of possible artifacts/issues...). Figure 1 is rather generic, it would be good to see visually step by step the data processing in a similar sequential order. I recommend then to anticipate the description of the devices at the beginning of the methodology section.

**Answer:**

Reflecting the referee's opinion, '2.1 Recording video containing rain streaks using infrared surveillance camera' was newly created in the methodology section and a supplementary explanation was added as follows.

(Revised Manuscript, Lines 70-92)

**2.1 Recording video containing rain streaks using infrared surveillance camera**

The surveillance camera records video. The video looks continuous, but it is also composed of discrete still images, so-called frames. The frequency of recording frames (i.e., acquisition rate) is called frames per second (fps). In other words, fps is how many images are taken per second for recording video. Another important factor in video recording is exposure time. Exposure time, also called shutter speed, refers to the time the camera sensor is exposed to light to capture a single frame. The real raindrops are close to a circle, but in a single image, the raindrops look like a streak. This is because raindrops move at a high speed during the exposure time. Therefore, the raindrops that

moved during the exposure time are visualized in the rain streaks in a single frame.

Fig. 2 shows an example of capturing a raindrop for a single frame. Here, only the raindrops in the front and rare that a little far from the point of focus are clearly visible, and objects that are more than a certain distance appear invisible. That is, the point where the focus is best is called the focus plane, and there is a range in which it can be recognized that objects are focused before and after the focus plane. The closest plane that can be considered to be in focus is called the near-focus plane, and the farthest plane is called the far-focus plane. This range is generally called depth of field (DoF). Ultimately, the rainfall intensity can be estimated based on the volume and raindrops in the DoF.

On the other hand, the description of the applied device and input data is described in Section 3 in the original manuscript (lines 211-215). In addition, an example of processing data sequentially following Figure 1 is already described in Section 4 in the original manuscript (lines 279-315).

Q2. I would like to better understand the concept of "dark conditions". I invite the authors to elaborate more and discuss accordingly the perspectives of this type of measurements. How does the performance continuously evolve in the transitions from dark to light and vice versa?

**Answer:**

In this study, the term 'dark condition' refers to the case of low illuminance. In other words, the dark condition is a condition in which raindrops cannot be captured by a general surveillance camera with visible light. For this reason, we considered recording raindrops in dark conditions using infrared (IR) mode. In the methodology section, we have added a description of this type of measurement (i.e., surveillance camera with IR mode).

**(Revised Manuscript, Lines 85-88)**

In this study, an infrared surveillance camera was considered under dark conditions. Here, the dark condition refers to a condition in which raindrops cannot be captured by a general surveillance camera with visible light. Infrared cameras emit near-infrared rays through an infrared emitter and receive the reflected light from the objects. Accordingly, there is an advantage in that raindrops that are invisible to the human eye can be detected.

Recording video in the daytime is affected by many factors (background, sunlight, moving objects, etc.). Several conditions are required to capture the raindrops. For example, there should be a wall or tree that can serve as a background, and the amount of illuminance should not be too strong. In particular, the difference in the amount of illuminance is expected to cause an extreme difference in the accuracy of estimating rainfall intensity. On the other hand,

infrared cameras used at night can be immune to the effects of these factors. Therefore, it is thought that there would be less uncertainty in the night mode, but the result could be markedly different depending on the observation sites. However, unfortunately, this study focused on the infrared mode when recording at night. The performance between day and night mode will be compared in our future study. Q3. The evaluation needs more data (more precipitation events). This will also help to better understand the differences at the tail of the distribution illustrated in the manuscript. At the present stage it is very hard to understand the potential and the error structure of this new measurement principle.

**Answer:**

As mentioned by the referee, a new case of precipitation analysis was added. By adding an analysis case different from the previously presented precipitation case, the possibility of utilizing existing public CCTV images is presented through the rainfall intensity and DSD (Drop Size Distribution) calculation method based on the observation data of the infrared surveillance camera newly proposed in this study.

(Revised Manuscript, Lines 402-445)

Fig. 13 illustrates the time series of the number concentration and  $D_m$  obtained from CCTV and PARSIVEL for case 2. In both CCTV and PARSIVEL observation data, the number concentration for a diameter between 0.5 mm and 1.5 mm had a value between 500 mm-1m-3 to 1,000 mm-1m-3, and there was no significant change in the number concentration with time.

The maximum diameter also consistently had a value close to about 3 mm, and the  $D_m$  was also similar to about 1.5 mm because the maximum particle diameter and the number concentration of 1 mm intermediate drop had similar values.

From 0100 LST to 0230 LST, the maximum particle diameter through CCTV was overestimated, resulting in a large value close to 4 mm. As a result, the Dm value increased significantly to more

than 2 mm. PARSIVEL data showed a sharp decrease in the number concentration of 1 mm particles at 0030 LST, and an increase in  $D_m$  under the influence of the sharply decreased number concentration. However, in the case of CCTV, only raindrops smaller than of 2 mm were observed at the time, and there was a difference in that Dm did not increase and was maintained.

---

## Author Response (AR1)

**Response Letter**

**Atmospheric Measurement Techniques**

AMT-2022-196

Title: Estimation of raindrop size distribution and rain rate with infrared surveillance camera in dark conditions

Author(s): Jinwook Lee et al.

MS type: Research article

Iteration: Revised submission

**Referee #1**

***Overall Comments***

This manuscript provides an interesting new method to estimate microstructural and bulk rainfall properties from a CCTV camera. The idea is intriguing and the topic is fully appropriate for the journal. The manuscript needs overall a major and mandatory revision, as detailed in the following comments.

**Answer:**

The authors appreciate the valuable comments. As suggested by the referee, all the comments were considered in the revision of the manuscript.

**Q1. Methodology section: the authors move forward to directly discuss and describe the algorithm. Important information on the measurement device is missing at this point, crucial for the scope of this journal. The reader at this point has the following questions: what are the technical characteristics of the camera? What are the actual input data? (description of the images, their resolution, acquisition rate, discussion of possible artifacts/issues…). Figure 1 is rather generic, it would be good to see visually step by step the data processing in a similar sequential order. I recommend then to anticipate the description of the devices at the beginning of the methodology section.**

**Answer:**

As suggested by the referee, '2.1 Recording video containing rain streaks using infrared surveillance camera' has been newly inserted in the methodology section and a supplementary explanation has been added.

(Revised Manuscript, Lines 73-92)

**2.1 Recording video containing rain streaks using infrared surveillance camera**

The surveillance camera records video. The video looks continuous, but it is also composed of discrete still images, so-called frames. The frequency of recording frames (i.e., acquisition rate) is called frames per second (fps). In other words, fps is how many images are taken per second for recording video. Another important factor in video recording is exposure time. Exposure time, also called shutter speed, refers to the time the camera sensor is exposed to light to capture a single frame. The real raindrops are close to a circle, but in a single image, the raindrops look like a streak. This is because raindrops move at a high speed during the exposure time. Therefore, the raindrops that

moved during the exposure time are visualized in the rain streaks in a single frame.

Fig. 1 shows an example of capturing a raindrop for a single frame. Here, only the raindrops near the point of focus are visible, and objects that are more than a certain distance appear invisible. That is, the point where the focus is best is called the focus plane, and there is a range in which it can be recognized that objects are focused before and after the focus plane. The closest plane that can be considered to be in focus is called the near-focus plane, and the farthest plane is called the far-focus plane. This range is generally called depth of field (DoF). Ultimately, the rainfall intensity can be estimated based on the volume and raindrops in the DoF.

In this study, an infrared surveillance camera was considered under dark conditions. Here, the dark condition refers to a condition in which raindrops cannot be captured by a general surveillance camera with visible light. Infrared cameras emit near-infrared rays through an infrared emitter and receive the reflected light from the objects. Accordingly, it has the advantage of being able to detect raindrops that are invisible to the human eye.

[Figure]

**Figure 1: Schematic diagram of the photographed rain streak in the image and the movement of a raindrop during the exposure time.**

On the other hand, the description of the applied device and input data is described in Section 3 in the original manuscript (lines 211-215). In addition, an example of processing data sequentially following the flow chart (Fig. 1, Original manuscript) is already described in Section 4 in the original manuscript (lines 279-315).

**Q2. I would like to better understand the concept of "dark conditions". I invite the authors to elaborate more and discuss accordingly the perspectives of this type of measurements. How does the performance continuously evolve in the transitions from dark to light and vice versa?**

**Answer:**

In this study, the term 'dark condition' refers to the case of low illuminance. In other words, the dark condition is a condition in which raindrops cannot be captured by a general surveillance camera with visible light. For this reason, we considered recording raindrops in dark conditions using infrared (IR) mode. In the methodology section, we have added a description of this type of measurement (i.e., surveillance camera with IR mode).

(Revised Manuscript, Lines 87-90)

> In this study, an infrared surveillance camera was considered under dark conditions. Here, the dark condition refers to a condition in which raindrops cannot be captured by a general surveillance camera with visible light. Infrared cameras emit near-infrared rays through an infrared emitter and receive the reflected light from the objects. Accordingly, it has the advantage of being able to detect raindrops that are invisible to the human eye.

Recording video in the daytime is affected by many factors (background, sunlight, moving objects, etc.). Several conditions are required to capture the raindrops. For example, there should be a wall or tree that can serve as a background, and the amount of illuminance should not be too strong. In particular, the difference in the amount of illuminance is expected to cause an extreme difference in the accuracy of estimating rainfall intensity. On the other hand,

infrared cameras used at night can be immune to the effects of these factors. Therefore, there would be less uncertainty in the night mode, but the result could be markedly different depending on the observation sites. However, unfortunately, this study focused on the infrared mode when recording at night. The performance between day and night modes will be compared in our future study.

**Q3. The evaluation needs more data (more precipitation events). This will also help to better understand the differences at the tail of the distribution illustrated in the manuscript. At the present stage it is very hard to understand the potential and the error structure of this new measurement principle.**

**Answer:**

As suggested by the referee, a new case of precipitation analysis has been added. By adding an analysis case different from the previously presented precipitation case, it can help to understand the overestimation tendency for drops larger than 3 mm in the drop size distribution estimated based on CCTV image and the underestimation characteristics of raindrops smaller than 1 mm in the PARSIVEL observation data.

In addition, CCTV observation is significant in that the dependence of number concentration on the channel size of drop size disdrometers can be reduced. While the PARSIVEL disdrometer only considers the observed diameter in estimating the raindrop size, the estimated raindrops from rain streaks in the CCTV image should also consider the actual drop size of the pixel and the distance from the camera lens. The distance from the lens could have some errors. Furthermore, the number concentration of raindrops can be overestimated due to the overlapping rain streaks on the CCTV image. To reduce the overestimation tendency, a method of calculating the short diameter through the central axis information of the rain streak was proposed. The authors have described the contents in the revised manuscript.

[revised manuscript text omitted]

**Q4. I do not see any statement about data and code availability. I strongly recommend to provide the data as well as the code in an appropriate repository. I consider it almost mandatory for this type of papers describing new methods.**

**Answer:**

The authors agree with the referee's opinion. The analysis of this study was performed through the collaboration of the authors. Thus, codes are written in multiple languages in IDL, Matlab, and Python. The partial code and data will be provided upon request of the readers, and this code availability was mentioned in the revised manuscript. In addition, the authors plan to integrate all codes into the python language and share them through Github.

(Revised Manuscript, Lines 466-467)

**Data availability**

The data and code can be provided by the corresponding author (hjkim22@cau.ac.kr) upon request.

**Q5. L12: please quantify "similar"**

**Answer:**

The numerical difference in the number concentration obtained through CCTV and PARSIVEL observations for the 0.5 - 1.5 mm diameter section was presented, indicating that the number concentration values were similar.

(Revised Manuscript, Lines 11-14)

Second, the number concentration of raindrops obtained through closed-circuit television (CCTV) images had values between 100 $mm^{-1}m^{-3}$ and 1,000 $mm^{-1}m^{-3}$, the RMSE for the number concentration by CCTV and PARticle Size and VELocity (PARSIVEL) was 72.3 $mm^{-1}m^{-3}$ and 131.6 $mm^{-1}m^{-3}$  in the 0.5 to 1.5 mm section.

**Q6. L13: it is not clear why you focus here only on the 0.5 to 1.5 mm interval**

**Answer:**

The rain rate calculated through the raindrop size distribution is proportional to the third power of the diameter of the raindrops, and the difference in the influence weight appears according to the diameter difference. Reference Figure 6-1 shows the contribution rate to rain rate by diameter for each rain rate section. The precipitation cases selected in this study mainly focus on weak rainfall of less than 5 mm h$^{-1}$. The rain rate value in the rain rate section weaker than 5 mm h$^{-1}$ is affected by the number concentration value for the 0-2 mm diameter section by more than 80% and as large as more than 90%. The high accuracy of number concentration obtained from CCTV image data in the corresponding diameter section affects the accuracy of rain rate estimation.

In addition, as shown in the PARSIVEL correction factor (Fig. 6-2) proposed by Raupach and Berne (2015), it can be seen that the difference in number concentration based on PARSIVEL and 2DVD (Two-dimensional Video Disdrometer) observation data in less than 0.5 mm diameter section increases sharply compared to the number concentration difference in the 1 mm. This result means that the accuracy of the number concentration value based on the PARSIVEL observation data for drops smaller than 0.5 mm is low. Therefore, this study focused on the 0.5-1.5 mm diameter section considering the constraints of the PARSIVEL observation instrument and the effect on the rain rate calculation accuracy.

[Figure]

**Fig. 6-1. The contribution rate of the N(D) of each diameter category to the rain rate (Kim et al., 2022).**
*(R1: ~ 1 mm h⁻¹, R2: 1~5 mm h⁻¹, R3: 5~10 mm h⁻¹, R4: 10~20 mm h⁻¹)*

[Figure]

**Fig. 6-2. Median P(i) values classed by Parsivel-derived intensity (Raupach and Berne, 2015).**

Reflecting 'correction factors' for PARSIVEL observation, the authors have modified the

related contents.

(Revised Manuscript, Lines 267-272)

We considered two rainfall events from 1945 LST on March 25, 2022, to 0615 LST on March 26, 2022 (case 1), and 2100 LST on September 5, 2022, to 0300 LST on September 5, 2022 (case 2). Fig. 6 illustrates the hyetographs of the rainfall event considered in this study according to the time resolution. The total rainfall of case 1 and 2 is 19.5 and 48.7 mm based on the PARSIVEL, respectively. The maximum rain rate is 10.0 and 20.7 mm h⁻¹ based on the 1 min resolution, and 5.0 and 14.5 mm h⁻¹ based on the 15 min resolution for case 1 and case 2.

[Figure]

(a) Case 1

[Figure]

(b) Case 2

**Figure 6: Hyetograph of PARSIVEL and rain gauge observation data for the rainfall events considered in this study (left: 1 min resolution, right: 10 min resolution).**

**References**

Kim, H. J., Jung, W., Suh, S. H., Lee, D. I., & You, C. H. (2022). The Characteristics of Raindrop Size Distribution at Windward and Leeward Side over Mountain Area. Remote Sensing, 14(10), 2419.

Raupach, T. H., & Berne, A. (2015). Correction of raindrop size distributions measured by Parsivel disdrometers, using a two-dimensional video disdrometer as a reference. Atmospheric Measurement Techniques, 8(1), 343-365.

**Q7. L20-25: please note that weighing gauges are nowadays used very often instead of tipping bucket.**

**Answer:**

As suggested by the referee, some descriptions related to the weighing gauges and comparison with the tipping-bucket type have been added.

(Revised Manuscript, Lines 26-30)

For this reason, weighing gauges are nowadays used very often instead of tipping-bucket-type. the weighing gauge is a meteorological instrument used to observe and analyze various precipitation, including rainfall and snowfall. Also, the tipping bucket has a large error due to the observation time delay when the rainfall is less than 10 mm h$^{-1}$ compared to the weighing gauge. However, when the observation time size is set to 10 to 15 minutes, the relative percentage error has a very low value of -6.7 - 2.5%, resulting in high accuracy (Colli et al., 2014).

(Newly added reference)

Colli, M., Lanza, L. G., La Barbera, P., Chan, P. W.: Measurement accuracy of weighing and tipping-bucket rainfall intensity gauges under dynamic laboratory testing. Atmos. Res., 144, 186-194, 2014.

**Q8. L63: provide a reference for the PARSIVEL instrument**

**Answer:**

As suggested by the referee, the reference for the PARSIVEL instrument has been added.

(Revised Manuscript, Lines 70-71)

The DSD was used to calculate rainfall intensity with physical optics analysis and verified using a PArticle SIze and VELocity (PARSIVEL) disdrometer (Löffler-Mang and Joss, 2000).

(References - already cited)

Löffler–Mang, M., Joss, J.: An optical disdrometer for measuring size and velocity of hydrometeors. J. Atmos. Ocean. Technol. 17 (2), 130–139, 2000.

**Q9. Equation 5: please note that there may be significant uncertainties to this relation. I suggest a discussion about it after revisiting the relevant literature on the subject.**

**Answer:**

The authors appreciate the detailed comment. Eq. (5) is the terminal velocity relationship for each diameter of raindrops. Marzuki et al. (2013) analyzed the fall velocity distribution by raindrop diameter for various rainfall types using long-term two-dimensional video disdrometer (2DVD) observation data. The precipitation cases they selected were precipitation with winds in a wide range of 0.5 to 9 m s$^{-1}$, and all precipitation cases reported by Atlas et al. (1973) suggested that it has a terminal velocity distribution. The precipitation cases they selected were precipitation cases accompanied by winds in a wide range of 0.5 to 9 m s$^{-1}$, all precipitation cases are fitted by terminal velocity distribution suggested by Atlas et al. (1973). In addition, as shown in Figure 9-1(b), even in the case of precipitation accompanied by strong wind speed, the difference between the fall velocity and the terminal velocity of the raindrops smaller than 4 mm was very low, less than 0.4 m s$^{-1}$.

[Figure]

**Fig. 9-1. (a) Fall velocity distribution by raindrop diameter, (b) average distribution of the difference between fall velocity and terminal velocity (Marzuki et al, 2013).**

Chen et al. (2022) compared and presented the terminal velocity distribution by raindrop diameter based on observation and theoretical relationships. As shown in Figure 9-2, the difference in the fall velocity of raindrops with a diameter of 0 to 4 mm did not appear significantly. During precipitation, the diameter of most of the raindrops is usually less than 3 mm, so it can be seen that the terminal velocity relationship is followed except for exceptional cases such as typhoons.

[Figure]

**Fig. 9-2. Distribution of terminal velocity by diameter of raindrops based on various types of models (Chen et al, 2022).**

**References**

Marzuki, Randeu, W. L., Kozu, T., Shimomai, T., Hashiguchi, H., & Schönhuber, M. (2013). Raindrop axis ratios, fall velocities and size distribution over Sumatra from 2D-Video Disdrometer measurement. Atmospheric Research, 119, 23-37.

Chen, J. P., Hsieh, T. W., Lin, Y. C., & Yu, C. K. (2022). Accurate parameterization of precipitation particles' fall speeds for bulk cloud microphysics schemes. Atmospheric Research, 273, 106171.

**Q10. Equation 8 (and discussion): is it possible also to obtain non-parametric (histograms) DSDs with this instrument? I would be curious to see how such histograms would look like.**

**Answer:**

This manuscript describes the analysis results based on the results of non-parametric DSDs based on actual observation data. The non-parametric DSDs, and average number concentration distribution by diameter can be found in Fig. 10 and Fig. 14 in the revised manuscript.

(Revised Manuscript, Fig. 10 and Fig. 14)

[Figure]

**Figure 109: Average number concentration versus diameter from the surveillance camera images and the PARSIVEL.**

[Figure]

**Figure 14: Average number concentration versus diameter from the surveillance camera images and the PARSIVEL.**

**Q11. L108: here the depth of field is mentioned. However, it was not previously introduced and discussed. See my larger comment on the methodology section.**

**Answer:**

As suggested by the referee, '2.1 Recording video containing rain streaks using infrared surveillance camera' was newly created in the methodology section, and a supplementary explanation for depth of field (DoF) was added.

(Revised Manuscript, Lines 81-90)

Fig. 1 shows an example of capturing a raindrop for a single frame. Here, only the raindrops near the point of focus are visible, and objects that are more than a certain distance appear invisible. That is, the point where the focus is best is called the focus plane, and there is a range in which it can be recognized that objects are focused before and after the focus plane. The closest plane that can be considered to be in focus is called the near-focus plane, and the farthest plane is called the far-focus plane. This range is generally called depth of field (DoF). Ultimately, the rainfall intensity can be estimated based on the volume and raindrops in the DoF.

In this study, an infrared surveillance camera was considered under dark conditions. Here, the dark condition refers to a condition in which raindrops cannot be captured by a general surveillance camera with visible light. Infrared cameras emit near-infrared rays through an infrared emitter and receive the reflected light from the objects. Accordingly, it has the advantage of being able to detect raindrops that are invisible to the human eye.

[Figure]

**Figure 1: Schematic diagram of the photographed rain streak in the image and the movement of a raindrop during the exposure time**

**Q12. Table 2 and Table 3: I would recommend to move this information to the Appendix.**

**Answer:**

As suggested by the Referee, Table 2 and Table 3 have been moved to the Appendix section.

**Q13. Figure 5: OK to show the data with different granularity, but I would like to see also the two time series with the same temporal resolution (by aggregating PARSIVEL data) as well as their cumulative curves, to understand if the Parsivel and the gauge are in decent agreement. Also, Figure 10 later on should be replicated to compare, at 15 minutes, the CCTV and the rain gauge which remains the real reference for rainfall amounts.**

**Answer:**

The authors apologize for the confusion in the review process. The content related to the rain gauge should have been deleted, but it was entered by mistake during the editing process by the authors. At first, we tried to include analysis related to the rain gauge as well as PARSIVEL. But this study was changed to intend focus on comparison with PARSIVEL data because there were many missing data from the rain gauge. Therefore, the content related to the rain gauge has been removed in the revised manuscript as follows. The contents mentioned by the referee will be actively reflected in our future research.

(Revised Manuscript, Line 10)

(Revised Manuscript, Line 467)

**Q14. Figure 8: the labels (a) and (b) are missing**

**Answer:**

The authors appreciate the thoughful comment. However, you can see the labels (a) and (b) below the number concentration time series distribution in Figure 8 in original manuscript (Revised Manuscript, Figure 9). Please check.

**Q15. Figure 9 (and discussion): why do you need to fit a gamma distribution for the Parsivel? Could you just use the non parametric form from the measurements?**

**Answer:**

The authors used a gamma distribution just for comparison with the results of non-parametric DSDs obtained from actual observation data. The rain rate was estimated using the information on non-parametric DSDs.

When the technique proposed in this study is applied to public CCTV, differences in the number concentration distribution may appear depending on various hardware conditions, image resolution, and differences in the observation area. Therefore, it is possible to estimate the rainfall intensity in which non-ideal drops are filtered by estimating the concentration distribution after obtaining the size and distribution information of the distribution of raindrops obtained in the image.

In addition, if the accuracy of estimating DSDs through CCTV images is secured, it is possible to acquire DSDs observation data with high observation resolution in various areas. The collected regional DSDs can be expected to increase their utility, such as verifying ground rainfall for microphysical schemes of numerical models.

**Q16. Table 5 (and discussion): I believe you should increase the size of your side-by-side comparison dataset. One rainfall event is not enough in my opinion.**

**Answer:**

As suggested by the referee, analysis for a case of rainfall event has been added to allow a comparison of results for the existing rainfall case. By presenting the results of different rainfall cases and the accuracy of rainfall intensity estimation through this study, it would be possible to use it as a reference for future CCTV video-based rainfall estimation research.

**Q17. The Parsivel has its own limitations. How were the data corrected or processed in order to be sure of its measurements to be taken as reference? (example https://doi.org/10.5194/amt-8-343-2015 but other relevant literature on Parsivel data processing is available)**

**Answer:**

The authors appreciate the referee's very detailed review. The authors think that the quality of this study could be improved through the referee's opinions. The number concentration correction factor of the PARSIVEL disdrometer observation data proposed by Raupach and Berne (2015) was applied to the actual observation data, and the related result figure was modified. Figure 17-1 compares the number concentration distribution before and after applying the correction factors. As a result of applying the correction factors of the number concentration, the number concentration value for the diameter section of 0.2 to 1 mm became smaller, so it can be seen that the difference with the number concentration distribution calculated through CCTV image data is reduced.

[Figure]

(a) PARSIVEL (non-corrected)  (b) PARSIVEL (corrected)

(c) CCTV

**Fig. 17-1. Time series of number concentration based on (a) PARSIVEL (non-corrected), (b) PARSIVEL (corrected), (c) CCTV observation data.**

**\* References**

Raupach, T. H., & Berne, A. (2015). Correction of raindrop size distributions measured by Parsivel disdrometers, using a two-dimensional video disdrometer as a reference. Atmospheric Measurement Techniques, 8(1), 343-365.

Reflecting on the number concentration correction factor of the PARSIVEL observation, the authors have modified the related contents.

(Revised Manuscript, Lines 267-272)

We considered two rainfall event from 1945 LST on March 25, 2022, to 0615 LST on March

26, 2022 (case 1), and 2100 LST on September 5, 2022, to 0300 LST on September 5, 2022 (case 2). Fig. 6 illustrates the hyetographs of the rainfall event considered in this study according to the time resolution. The total rainfall of case 1 and 2 is 19.5 and 48.7 mm based on the PARSIVEL, respectively. The maximum rain rate is 10.0 and 20.7 mm h$^{-1}$ based on the 1 min resolution, and 5.0 and 14.5 mm h$^{-1}$ based on the 15 min resolution for case 1 and case 2.

(a) 1 min          (b) 15 min

(a) Case 1

36 LST

[Figure]

(b) Case 2

**Figure 6: Hyetograph of PARSIVEL and rain gauge observation data for the rainfall events considered in this study (left: 1 min resolution, right: 10 min resolution).**

**Q18. Figure 10 (and discussion): please comment more in -depth about the origin of the extremely large overestimations around 20 LST and 06 LST. I am interested to see exactly how the transition from light to dark affects the data.**

**Answer:**

The rainfall event was taken in infrared mode under dark conditions. Therefore, we think it is safe to say that the transition effect from light to dark is almost negligible. However, large overestimation around 20 LST and 06 LST seems to have occurred for the following reasons. 1) When there is a lot of rainfall, there are many overlapping rain streaks in an image. 2) The current algorithm recognizes overlapping rain streaks as a single big raindrop. 3) Large-scale raindrops are overestimated in drop size distribution. 4) Rainfall intensity is overestimated. A similar tendency was confirmed in the newly added rainfall event.

However, some modifications were made to the algorithm to alleviate the overestimation problem during the revision process of the manuscript. Based on this, the problem of overestimation could be improved a little. However, it is still difficult to completely solve this overestimation problem caused by overlapping raindrops. The authors have added a discussion related to this issue.

(Revised Manuscript, Lines 311-314)

If the rain streaks overlap, the diameter of the raindrops can be estimated as large. To reduce the overestimation of raindrop diameter, this study tried to find the main central axis

coordinates of overlapping rain streaks and set the longest central axis as the representative value. Then, estimate the primary diameter by calculating the distance between each pixel value of the set central axis and the edge pixels of rain streaks.

(Revised Manuscript, Lines 426-427)

This tendency was confirmed in case 1 which may be caused by recognizing overlapping rain streaks as a single big raindrop.

**Referee #2**

Opportunistic sensing is an emerging crowdsensing technique for monitoring precipitation. Previous studies have suggested that delicate use of visual surveillance cameras allow the retrievals of rain drop size distributions as well as rainfall intensity. This study demonstrates that raindrop size distributions can be retrieved from an infrared surveillance camera as well. The topic is relevant for AMT readers, and the presented work is interesting. I have a few concerns as listed below.

**Answer:**

The authors appreciate the valuable comments. As suggested by the referee, all the comments were considered in the revision of the manuscript.

**Q1. The motivation of using infrared surveillance cameras is weak to me. Although no such work has been done, it does not necessarily mean that the presented work is promising in applications. Given many readers are in the meteorology community, they may wonder: Are infrared surveillance cameras widely distributed? Why and how should this approach be applied? At what conditions should we employ this technique? The authors may elaborate this point in Introduction or in Discussions.**

**Answer:**

The surveillance camera (supporting infrared mode) considered in this study will not be able to replace traditional observation devices (rain gauge, radar, satellite, etc.). However, if these studies can be continued to secure robustness, the surveillance camera will be an excellent complement to the existing observation system in terms of spatiotemporal resolution and accuracy improvement.

The infrared surveillance camera mentioned in this study refers to a product in which an IR (infrared) light is additionally installed on a general camera. Infrared cameras emit near-infrared rays through an infrared emitter and receive the reflected light from the objects. Accordingly, there is an advantage in that raindrops, invisible to the human eye, can be detected. There are not many surveillance cameras that support infrared mode yet, most surveillance cameras currently installed have these functions and are widely distributed. For example, in Korea, where the authors reside, the most frequently installed public surveillance camera model is Hanhwa Techwin SNO-6084R, which supports IR mode (Please see, https://product.hanwha-security.com/en/products/camera/network/bullet/SNO-6084R/overview/#).

The reasons for using the proposed method are as follows. Existing studies have focused on the time when video can be captured with visible light. Naturally, these methodologies are only applicable in daytime conditions. In other words, it is impossible to obtain input data without visible light using the existing image-based rainfall measurement method. However, when photographing using infrared rays, it is possible to obtain a rainfall image even when there is no general sunlight. Therefore, it is of great significance in terms of supplementing the limitations of existing studies.

Conditions and manners for using the proposed methodology are as follows. The method can be used under low illumination conditions. That is, when the amount of light is sufficient, the video may be sufficiently captured with visible light, and thus the IR mode is not required. However, when sunlight is insufficient, an image should be captured using a camera supporting the IR mode. If the IR mode is not supported, it is also possible to additionally install IR light to the normal camera to record the video under dark conditions. When using a camera that supports IR mode, it can be set to the auto mode provided by the camera manufacturer, or the operating time zone can be set to manual. In general, this auto mode has the sensitivity to illumination as an option (Algorithms for auto mode are usually not disclosed to users). Therefore, if this sensitivity is well adjusted through trial and error, it is possible to acquire rainfall images under dark conditions. The manuscript was revised by reflecting the above-mentioned contents.

(Revised Manuscript, Lines 64-67)

However, the existing studies have focused on the time when video can be captured with visible

light. It is impossible to obtain input data without visible light using the existing image-based rainfall measurement method. Thus, these methodologies are only applicable in daytime conditions. However, when recording using infrared rays, it is possible to obtain a rainfall image even when there is no sunlight. No study has estimated the rain in dark conditions to our knowledge.

(Revised Manuscript, Lines 87-90)

In this study, an infrared surveillance camera was considered under dark conditions. Here, the dark condition refers to a condition in which raindrops cannot be captured by a general surveillance camera with visible light. Infrared cameras emit near-infrared rays through an infrared emitter and receive the reflected light from the objects. Accordingly, it has the advantage of being able to detect raindrops that are invisible to the human eye.

(Revised Manuscript, Lines 458-462)

In this study, DSD was estimated using an infrared surveillance camera; the rain rate was also estimated. Consequently, we could confirm the possibility of estimating an image-based DSD and rain rate obtained based on low-cost equipment in dark conditions. Though, the infrared surveillance camera considered in this study will not be able to replace traditional observation devices, if future studies can be continued to secure robustness, it will be an excellent complement to the existing observation system in terms of spatio-temporal resolution and accuracy improvement.

**Q2. It appears to me that the algorithm used in this study is similar with previous works on visual images. The authors should clearly state the innovative point of the presented algorithm. For example, how were the previous algorithms adapted to fit the infrared application?**

**Answer:**

Existing image processing-related studies regarded rainfall as noise that should be removed. However, recent studies, including this study, attempted to estimate rainfall using these noises. These studies used rainfall images to calculate rainfall intensity through techniques such as AI. However, unlike previous studies, individual rainfall particles were extracted through optical physics and image processing, and their distribution was compared with actual observation data from the PARSIVEL.

In addition, the algorithm presented in this study has innovative points adapted to fit the infrared application compared with the previous studies. Largely, 1) 2D kernel is additionally applied to improve the light smudging phenomenon that may occur in infrared video, and 2) only major raindrop is extracted for overlapping raindrops. The manuscript has been modified to reflect these parts.

(Revised Manuscript, Lines 137-138

Videos from infrared mode have usually a blur effect. Thus, the additional 2D kernel was applied to remove the pixels having blur.

(Revised Manuscript, Lines 311-314)

If the rain streaks overlap, the diameter of the raindrops can be estimated large. In this study, in order to reduce the overestimation of raindrop diameter, find the main central axis coordinates of overlapping rain streaks and set the longest central axis as the representative value. Then, estimate the primary diameter by calculating the distance between each pixel value of the set central axis and the edge pixels of rain streaks.

**Q3. Fig. 7. Where are those big particles from? If they are falling, they should have rather high velocities. But they could also be the results of lens contamination.**

**Answer:**

The possibility of lens contamination mentioned by the referee is a very important part of analysis using camera image data. The authors were aware of this and used two methods as countermeasures. The first method is to fundamentally block lens contamination. In other words, it is to install a shield on the surveillance camera itself. The next method is to take only an image of an uncontaminated part, and this meaningful part of the image is called a region of interest (ROI). The camera's resolution is 1,080 pixels for the height and 1,920 pixels for the width, but the images were cropped to images with 640×640 pixels applying ROI to get clean images.

In addition, the big raindrops described in the manuscript result from a large diameter estimation due to the overlapping rain streaks in the CCTV image. In order to reduce the overestimation of the raindrop diameter, the central axis coordinates of the rain streaks in the CCTV image were obtained, and the main diameter information was found based on the central axis information of the overlapping rain streaks. Based on this, the problem of overestimation could be improved a little. However, it is still difficult to completely solve this overestimation problem caused by overlapping raindrops. The authors have added a discussion related to this issue.

(Revised Manuscript, Lines 311-314)

If the rain streaks overlap, the diameter of the raindrops can be estimated as large. To reduce the overestimation of raindrop diameter, this study tried to find the main central axis coordinates of overlapping rain streaks and set the longest central axis as the representative value. Then, estimate the primary diameter by calculating the distance between each pixel value of the set central axis and the edge pixels of rain streaks.

(Revised Manuscript, Lines 426-427)

This tendency was confirmed in case 1 which may be caused by recognizing overlapping rain streaks as a single big raindrop.

**Q4. Fig. 8. Comparing the DSDs retrieved from the camera and PARSIVEL, It appears that the variation of DSDs is not well captured by the camera. In particular, significant overestimation has been found for large raindrops. The contributing factors should be discussed.**

**Answer:**

Since the PARSIVEL data also has an error in the observation data of the in-situ instrument, the variance was reduced by applying the correction factor for each diameter proposed in previous research results (Raupach and Berne, 2015). However, the problem of underestimating the number concentration for raindrops smaller than 0.5 mm with the PARSIVEL disdrometer still existed. In order to reduce the problems of these instruments, we plan to conduct comparative verification with 2DVD (Two-Dimensional Video Disdrometer), which has higher observation accuracy for small rain particles of less than 0.5 mm compared to PARSIVEL.

Large raindrops can be seen as occurring in calculating raindrop diameter based on rain streak information by overlapping rain streaks. The length of the rain streak is highly dependent on the camera's exposure time, and as the exposure time decreases, the length of the rain streak becomes shorter. As a result, the overlapping phenomenon of rain streaks can be reduced. This study focuses on increasing the meteorological usability of existing CCTV image data, not on instruments produced for research. Future research results related to hardware settings, such as exposure time and shutter speed, will be performed to suggest ways to utilize CCTV data for disaster prevention. The authors have added a discussion related to the contributing factors of overestimation.

(Revised Manuscript, Lines 426-427)

This tendency was confirmed in case 1 which may be caused by recognizing overlapping rain streaks as a single big raindrop.

**\* References**

Raupach, T. H., & Berne, A. (2015). Correction of raindrop size distributions measured by Parsivel disdrometers, using a two-dimensional video disdrometer as a reference. Atmospheric Measurement Techniques, 8(1), 343-365.

**Q5. Fig. 9. It appears that fitting a distribution to some extent alleviates the overestimation of large drop concentrations, have you tried to construct the DSD using the fitted distribution? I would expect improved results.**

**Answer:**

As noted by the referee, fitting distributions based on observational DSD can lower the number concentration variance by diameter. Rain rate estimation accuracy can be improved through fitting distribution information, and rainfall spatial distribution accuracy of remote sensing data such as satellites and radars may be improved by using CCTV image data. However, the fitting distribution has a high dependence on the fitting model, which can cause errors by diameter, and the density dependence of small raindrops is also high. By improving the observation data's accuracy, the fitting distribution's accuracy can be expected to be improved. Therefore, this study focuses on the raindrop diameter estimation method observed in video image data. The method for estimating the diameter of raindrops based on the central axis information of rain streaks was improved. The improved results for diameter overestimation due to overlapping rain streaks were additionally described in the manuscript.

(Revised Manuscript, Lines 311-314)

If the rain streaks overlap, the diameter of the raindrops can be estimated as large. To reduce the overestimation of raindrop diameter, this study tried to find the main central axis coordinates of overlapping rain streaks and set the longest central axis as the representative

value. Then, estimate the primary diameter by calculating the distance between each pixel

value of the set central axis and the edge pixels of rain streaks

(Revised Manuscript, Figure 11)

[Figure]

**Figure 11: The rain rate time series calculated from the surveillance camera images (gray bar) and PARSIVEL observation data (red line) from 1945 LST on March 25 to 0600 LST on March 26, 2022.**

**Q6. Given the significant bias found for large raindrops, I believe the evaluation should be made for a heavy rainfall event. Otherwise, the story is incomplete.**

**Answer:**

We agree with the referee's opinion. By adding an analysis case, the authors tried to increase the reliability of the research results for rain rate and DSD (Drop Size Distribution) estimation based on general CCTV image data, which is the aim of this research.

---

## Author Response (AR2)

**Response Letter**

**Atmospheric Measurement Techniques**

AMT-2022-196

Title: Estimation of raindrop size distribution and rain rate with infrared surveillance camera in dark conditions

Author(s): Jinwook Lee et al.

MS type: Research article

Iteration: Revised submission

**Referee #1**

**Overall Comments**

I appreciated the revision efforts of the authors and I still believe the article is relevant for AMT.

I have the following concerns that I would like to see addressed prior to publication.

**Answer:**

The authors appreciate the valuable comments. As suggested by the referee, all the comments were considered in the revision of the manuscript.

**Q1. I would like the limitations of this method to be better stated at the beginning of the paper, especially in terms of scalability and overall relevance of the approach for meteorological measurements. The methodology is technically correct and interesting, but still limited in terms of applicability (even foreseen potential applicability) for what can be seen in this manuscript. The limitation to night-time measurements should be further stressed.**

**Answer:**

The authors appreciate the valuable comment. As suggested by the referee, we have modified the manuscript to better state the limitations of the method at the beginning of the paper.

(Revised Manuscript (Track), Lines 58-71)

It is true that these previous studies confirmed the possibility of rainfall measurement using surveillance cameras. However, several limitations still prevent the actual expansion of the measurement systems using surveillance cameras. In general, most surveillance cameras are installed for monitoring purposes, and people's faces are inevitably captured. Therefore, it is not easy to disclose the data due to privacy concerns. Data storage and transmission are also limitations. Since most surveillance cameras use a hard disk, data must be taken out directly. In other words, rainfall estimation cannot be done in real-time unless a system is in place to transmit data over the Internet. In addition, the applicability to night-time is more limited. In the case of general surveillance cameras in the past, observation is possible only when sunlight exists. For the observation system to expand, these various limitations must be addressed, and it seems that a lot of time and effort are needed. Nevertheless, research to develop algorithms using surveillance cameras in various conditions and to

confirm applicability can have sufficient meaning. The case of dark conditions is one of the conditions worth studying. This is because the recently installed surveillance cameras are equipped with an Infrared recording function, so most cameras will be able to take videos at night soon. However, the final purpose of utilizing these devices and the method is not to replace existing devices. It could be a supplement to improve the spatiotemporal resolution and accuracy of existing observation instruments. In particular, a study on the drop size distribution of rainfall, rather than simple rainfall estimation, would have more potential application value.

**Q2. If there was a rain gauge on-site, even if not working 100% of the time, it is crucial in my view that it is used to show some evaluation with respect to rainfall rate estimation. Otherwise the reader will always have the legitimate doubt whether the PARSIVEL is actually a good reference for rainfall rate at the range of rainfall rates shown in the study. I suggest to reconsider the decision to remove the rain gauge from the analysis.**

**Answer:**

The authors agree with the referee's comment. As the referee mentioned, securing the quantitative reliability of observational data is very important. Therefore, the authors have added the contents of quantitative verification with rainfall data obtained from rain gauge site operated by KMA (Korea Meteorological Administration) located close to the PARSIVEL observation site to the manuscript.

(Revised Manuscript (Track), Lines 301-311)

> In order to secure the quantitative reliability of the PARSIVEL observation data, rain gauge observation data were used to verify the rainfall calculated through the PARSIVEL observation. The rainfall data used for verification are rain gauge observation data operated by KMA (Korea Meteorological Administration) installed closer than 4 km from the PARSIVEL observation site (Table 3). The rainfall comparison period is from September 14, 2021, to October 4, 2022, including the period of the analysis case. Fig. 7 shows scatter plots comparing hourly rain rates from rain gauges and PARSIVEL. As a result of comparison with the observation data at three rain gauge sites, it had low MAE (Mean Absolute Error), RMSE (Root Mean Square Error), MAPE (Mean Absolute Percent Error) values of less than 0.11 mm h$^{-1}$, 0.6 mm h-1, and 8%. Also, correlation values were more than 0.9.

**Table 3. Location information of rain gauge observation sites.**

| Raingague site | Latitude (°) | Longitude (°) | Range from PARSIVEL site (km) |
|---|---|---|---|
| G1 | 37.4933 | 126.9175 | 3.73 |
| G2 | 37.5196 | 126.9763 | 2.42 |
| G3 | 37.5249 | 126.9390 | 2.87 |

(a) G1            (b) G2

(a) G3

**Figure 7. Scatter plot of rainfall amount every 1 hour from the PARSIVEL observation and the rain gauge observation.**

**Q3. Data and code availability: I find it a little weak for this paper to propose the "available upon request" approach. It is not sustainable in the long term. I recommend to have data and code in an appropriate repositories, as previously suggested.**

**Answer:**

The authors agree with the referee's comment. Suggest by the referee, the authors uploaded the videos, data, and sample codes used in the study to 'figshare' and 'GitHub' to increase the availability of the code and data used in this study.

(Revised Manuscript (Track), Lines 519-521)

> The raw videos and data used in the analysis can be downloaded from https://doi.org/10.6084/m9.figshare.c.6392430.v1, and the sample codes are available in a public GitHub repository from https://github.com/jinwook213/Rain_CCTV.git.